# Analytical Methods for the Determination of Diamorphine (Heroin) in Biological Matrices: A Review

**DOI:** 10.3390/toxics13100867

**Published:** 2025-10-13

**Authors:** Ahmed Ibrahim Al-Asmari

**Affiliations:** 1Special Toxicological Analysis Section, Pathology and Laboratory Medicine Division, King Faisal Special Hospital and Research Center, P.O. Box 3354, Riyadh 11211, Saudi Arabia; aalasmari1@kfshrc.edu.sa; 2Faculty of Medicine, Alfaisal University, Riyadh 11533, Saudi Arabia

**Keywords:** forensics, toxicology, pharmacokinetics, metabolites, opioids, opiates

## Abstract

Diamorphine (DIM, heroin) is a semi-synthetic opioid that undergoes rapid conversion to 6-monoacetylmorphine and morphine, producing short-lived biomarkers that are difficult to capture during the process. This review critically explores the evolution of analytical techniques for quantitative DIM analysis in biological matrices from 1980 to 2025. It synthesizes findings across blood, plasma, urine, hair, sweat, and postmortem samples, emphasizing matrix-specific challenges and forensic applicability. Unlike previous opioid reviews that primarily focused on metabolites, this work highlights analytical methods capable of successfully detecting diamorphine itself alongside its key metabolites. This review examines 32 studies spanning three decades and compares three core analytical methods: gas chromatography–mass spectrometry (GC–MS), high-performance liquid chromatography (HPLC) with optical detection and liquid chromatography–mass spectrometry (LC–MS). Key performance metrics include sensitivity, sample preparation workflow, hydrolysis control, metabolite coverage, matrix compatibility, automation potential and throughput. GC–MS remains the workhorse for hair and sweat ultra-trace screening after derivatization. HPLC with UV, fluorescence or diode-array detection enables robust quantification of morphine and its glucuronides in pharmacokinetic and clinical settings. LC–MS facilitates the multiplexed analysis of DIM, its ester metabolites and its conjugates in a single, rapid run under gentle conditions to prevent ex vivo degradation. Recent advances such as high-resolution mass spectrometry and microsampling techniques offer new opportunities for sensitive and matrix-adapted analysis. By integrating validation parameters, forensic applicability, and evolving instrumentation, this review provides a practical roadmap for toxicologists and analysts navigating complex biological evidence.

## 1. Introduction

Diamorphine (DIM), known colloquially as heroin, is a semi-synthetic opioid synthesized by acetylating the hydroxyl groups in morphine, a chemical transformation first achieved in 1874, leading to the first introduction of the drug into medicine in 1898 [1,2,3]. According to the United Nations Office on Drugs and Crime, DIM is the primary cause of most opioid-related fatalities worldwide [4].

Upon administration, DIM is rapidly metabolized to 6-monoacetylmorphine (6-MAM), a short-lived intermediate, and subsequently metabolized to morphine. While codeine is not a direct metabolite of DIM, it may appear in postmortem samples following the use of “street heroin,” which often contains 6-acetylcodeine (6-AC), a synthetic byproduct that converts to codeine in vivo [1,2]. The forensic detection of DIM poses immense analytical challenges. Key biomarkers, such as 6-MAM and 6-AC, have short half-lives (t½) and often persist only at trace levels, requiring highly sensitive and validated methods to confirm exposure [5,6]. The rapid degradation of DIM, especially in postmortem fluids or during delayed sample processing, can obscure its contribution to overdose and confound medico-legal interpretations [7,8,9].

The pharmacokinetic profile of DIM is dominated by an almost instantaneous deacetylation of 6-MAM, catalyzed by ubiquitous plasma and tissue esterases. This rapid turnover gives DIM a remarkably brief plasma t½, with studies reporting DIM t½ at just 2–8 min following intravenous (IV) administration [10]. Consequently, detecting unchanged DIM in blood typically requires sampling within minutes of injection or in the context of massive overdose, when circulating concentrations briefly rise above analytical limits [10,11,12].

The analytical challenges posed by DIM and its short-lived metabolites had been recognized as early as the 1980s, when Umans et al. [13] first employed high-performance liquid chromatography with ultraviolet detector (HPLC-UV)) in detecting DIM and its primary metabolites (6-MAM and morphine) in blood. This pioneering work laid the foundational framework for opioid toxicology. Shortly thereafter, Inturrisi et al. [14] extended this approach to characterize the pharmacokinetics of DIM in patients with chronic pain, demonstrating the relevance of HPLC-UV, not only in forensic analysis but also in clinical pharmacotherapy.

In the 1990s, the scope of DIM detection expanded beyond blood-based matrices. Goldberger et al. [15] introduced N-tert-butyl-dimethylsilyl-N-methyltrifluoroacetamide-derivatized hair testing (MBTFA), enabling long-term retrospective profiling of DIM exposure. Cone et al. [16] and Kintz et al. [17] pioneered non-invasive methodologies by using gas chromatography–mass spectrometry (GC–MS) to validate sweat-patch sampling for capturing cumulative drug excretion within 24–72 h in DIM maintenance populations. Their pivotal studies demonstrated the possibility of accurately identifying and quantifying DIM in alternative matrices, such as hair and sweat. Their breakthrough greatly expanded the forensic toxicology toolkit and enabled long-term surveillance in both legal and clinical contexts.

However, despite these early breakthroughs, many DIM deaths remained analytically unresolved due to limited sensitivity or unstable biomarkers. The evolution of high-throughput methods, particularly liquid chromatography–(tandem) mass spectrometry (LC–MS[/MS]) with cold-chain protocols and mixed-mode solid-phase extraction (SPE), marked a turning point. Studies by Katagi et al. [18], Klous et al. [19] and Rook et al. [20] demonstrated that sub-nanogram quantification of DIM, 6-MAM, morphine and glucuronide conjugates was possible, with run times of under 10 min. Their groundwork has recently been expanded on using accurate-mass (Orbitrap or time of flight methods) and microsample workflows, such as dried blood spots, which offer greater sensitivity, broader metabolite coverage and the ability for retrospective detection in challenging matrices [21,22,23].

Although numerous analytical methods have been described and applied in forensic and clinical toxicology for the quantification of DIM, a systematic, side-by-side evaluation of these methods has not previously been undertaken. Unlike prior opioid-related reviews that primarily address morphine and its metabolites, this work concentrates exclusively on DIM quantification. The goal is to objectively examine analytical procedures for diamorphine across multiple biological matrices with a particular focus on forensic applicability. This review differs from earlier opioid reviews by systematically comparing DIM-specific methods side-by-side, harmonizing validation parameters, and highlighting matrix-dependent challenges. Its novelty lies in consolidating sample preparation workflows, metabolite profiling, and instrumental platforms into a single forensic-oriented framework, thereby providing laboratories with practical guidance for method selection, optimization, and validation in both clinical and postmortem contexts. As methods for the quantification of morphine and its metabolites have been reviewed elsewhere [24,25,26], only analytical methods for the quantification of DIM are reviewed in this work. Furthermore, this review contextualizes and compares the major analytical technologies used in DIM and metabolite detection, including derivatized GC–MS, LC–MS(/MS) and high-performance liquid chromatography (HPLC) coupled with different detectors, such as UV, fluorescent detector (FLD) or micellar liquid chromatography–UV detector (MLC–UV), across critical dimensions of sensitivity, sample preparation, matrix compatibility, hydrolysis control, metabolite coverage and automation. By mapping historical foundations to emerging high-sensitivity platforms, this review offers forensic and clinical laboratories an evidence-based framework to select, optimize and validate workflows suited for DIM-related casework, rehabilitation monitoring and postmortem investigations.

## 2. Materials and Methods

This narrative review systematically identifies and compares analytical workflows for the direct quantification of diamorphine (DIM) and its metabolites in biological specimens. A structured literature search was conducted in PubMed and Scopus covering the period from January 1980 to July 2025, using Boolean combinations of the keywords “heroin,” “diamorphine,” “6-MAM,” “morphine,” and “glucuronide.”. From an initial pool of 543 records (PubMed n = 131; Scopus n = 412), duplicates and irrelevant studies were removed. Articles were excluded if they focused exclusively on environmental or non-biological matrices, lacked sufficient validation data, or described methods quantifying only morphine or its glucuronide conjugates without direct DIM measurement. After removal of duplicates and irrelevant records, 470 titles and abstracts were screened, of which 378 were excluded. Ninety-two full texts were sought for retrieval, with 15 not retrieved. Of the 77 reports assessed for eligibility, 32 studies met the inclusion criteria and were included in the review, as indicated in Figure 1.

Studies were included if they applied validated analytical methods to biological matrices such as blood, plasma, urine, saliva, hair, sweat, vitreous humor, or tissues, and if they addressed contexts including clinical pharmacokinetics, maintenance therapy monitoring, or postmortem casework. Both pharmaceutical-grade DIM and seized street samples were considered, irrespective of reported purity, to ensure the review reflects real-world analytical challenges.

For each eligible study, information was extracted on the target analytes, sample preparation and cleanup strategies, chromatographic and detection platforms, validation metrics (limits of detection (LOD) and quantification (LOQ), linearity, precision, accuracy, recovery and matrix effects) and study cohorts. This structured approach enabled a harmonized and critically appraise the evolution of GC–MS, HPLC–UV/FLD/diode-array detection [DAD] and LC–MS(/MS) for DIM analysis across nearly four decades.

## 3. Results

### 3.1. Analytical Methods

#### 3.1.1. Detection Approaches and Analytical Families

In this review of 32 peer-reviewed studies published between 1982 and 2025 (Table 1), GC–MS, HPLC with different detectors and LC–MS^n^ account for most DIM and metabolite assays performed. Their predominance reflects several interrelated challenges in measuring DIM and its first-generation metabolites in addition to phase two metabolites, particularly morphine glucuronides:

Ultra-trace detection of a highly labile analyte revealed that the two acetyl esters of DIM hydrolyze within minutes to 6-MAM and morphine and circulating DIM concentrations often fall into the sub-nanogram-per-milliliter (sub-ng/mL) range. At the beginning of the DIM detection and quantification process, GC–MS methods using derivatization reagents, such as MBTFA, trimethylsilyl (TMS) or propionic anhydride, to stabilize those esters was considered the best option at the beginning of 90s, achieving picogram-level LODs in hair and sweat [15,27]. The need for rapid pharmacokinetic (PK)/pharmacodynamic (PD) profiling in clinical and forensic applications demand time-resolved sampling over minutes to hours post-dose. Morphine polar metabolites were not often measured using GC–MS techniques, as determining total and free morphine levels is tedious. However, glucuronide metabolites are not always inactive, i.e., morphine-6-glucronide is considered an active metabolite and exhibit some analgesic properties related to DIM or morphine. Thus, scientists developed methods for separating morphine polar metabolites for further analyses [28], which were then introduced in the 80 s and facilitated PK analysis of DIM and metabolites [29]. In some of earlier DIM studies, 6-MAM was detected using GC–MS while morphine and its glucuronides were detected using HPLC with UV, FLD or DAD, thereby bypassing derivatization and enabling sub-10-min runs. Furthermore, HPLC facilitated the reliable quantification of morphine and its glucuronides for PK modeling, albeit at LOQs, 3–25 ng/mL than the standard for the more abundant metabolites [30,31,32,33].

**Table 1 toxics-13-00867-t001:** Procedures for the quantification of diamorphine and its metabolites in biological samples published in 1980–2025.

Analytes *	Sample Types	Sample Preparation	Analytical Method ##	LOD # /LOQ ** (ng/mL)	Linearity	Recovery (%)	%CV	Stability	Application	Ref.
DIM 6-MAM Morphine	Blood	Rapid freezing → LLE (toluene–n-butanol (7:3, *v*/*v*, Burdick & Jackson Labs., Muskegon, MI, USA) → Wash with 0.1 N H_2_SO_4_ → pH adjustment to 8–9.0 with 1 N NaOH → Twice 5 mL Toluene-butanol extraction → Centrifuge 7 min → Evaporate under nitrogen at 40 °C to dryness → Reconstitute in 260 µL methanol (Burdick & Jackson Labs., Muskegon, MI, USA) → Inject 200 µL into HPLC	HPLC System → Varian Model 8500 (Varian, Sunnyvale, CA, USA) → Detector: Varichrom VUV-10 (Varian, Sunnyvale, CA, USA) → → Injector: Model 7125 (Rheodyne, Berkeley, CA, USA) → Recorder: Model A-25 (Varian, Sunnyvale, CA, USA) → Column: LiChrosorb Si-60 (30 cm × 4 mm I.D., 5-µm particle size (E. Merck, Darmstadt, Germany)) → Analytical Separation Phase: acetonitrile–methanol (75:25, *v*/*v*) → Flow Rate: 80 mL/h → Temp. Control: 30 °C	LOD = 6 LOS = 12.5 ULOQ = 4000	12.5–200	88 94 92	2–4 3.4–5 3–3.5	Total DIM loss during extraction: 11.2 ± 0.36%	This is the first report of the detection of DAM in human blood	[13]
DIM 6-MAM Morphine	Blood	Recruit 11 patients with chronic pain (9 with cancer) → Administer DIM hydrochloride via 4 routes: intravenous injection, intravenous infusion, intramuscular injection, and oral dosing → Collect serial blood samples at multiple time points post-administration → Separate plasma and store appropriately for analysis → Prepare samples for HPLC analysis (as described in Umans et al. [13], without further procedural detail provided in this paper)	Analytes → DIM, 6-MAM, morphine Instrumentation → HPLC with UV detection (Varian Model 8500 (Varian, Sunnyvale, CA, USA), Detection Wavelength → 210 nm Quantification → DIM and 6-MAM: Rapidly detected post-injection, short half-life → Morphine: Detected later, with slower clearance → Pharmacokinetic parameters calculated: Cmax, Tmax, t½, AUC, clearance	LOD = 6 LOS = 12.5	12.5–200	29–94	<5.1	Total DIM loss during extraction: 11.2 ± 0.36%	Pharmacokinetic application	[14]
DIM, 6-MAM Morphine Codeine	Hair (20 DIM users)	Collect hair samples (~100 mg) → Cut into ~1 mm segments → Place in fritted reservoirs → Wash with 1 mL methanol (reservoirs ) (30 s vortex) → Dry hair → Add 1 mL methanol + internal standards (100 ng each of D_3_-morphine, D_3_-codeine, D_3_-6-AM) → Seal and incubate at 37 °C for 18 h with stirring → Evaporate methanol under nitrogen at 50 °C → Add 2 mL deionized water + saturated sodium bicarbonate (pH 8.4) (Fisher Scientific, Fair Lawn, NJ, USA) → Extract with 7 mL toluene–heptane–isoamyl alcohol (70:20:10) (Burdick & Jackson Labs., Muskegon, MI, USA) → Shake 15 min → Centrifuge → Transfer organic layer → Evaporate at 60 °C under nitrogen → Add 40 µL N-methyl-bis(trifluoroacetamide) (MBTFA, Pierce Chemical Co., Rockford, IL, USA) → Cap, vortex, heat at 60 °C for 20 min → Transfer to autosampler vial for GC–MS	Instrument → HP 5890A GC + 5970A MSD, (SIM mode) (Hewlett-Packard, Palo Alto, CA, USA) → HP 7673A ALS (autosampler, Hewlett-Packard, Palo Alto, CA, USA) → Column → HP-5 (25 m × 0.32 mm i.d., 0.17 µm film, Hewlett-Packard, Palo Alto, CA, USA) → Carrier Gas → Helium at 2.5 mL/min Injection → Splitless, 250 °C inlet, 280 °C transfer line Oven Program → 200 °C (1 min) → Ramp 10 °C/min to 280 °C → Hold 3 min Detection → SIM of derivatized ions: → DIM: *m*/*z* 327 (quant), 204, 268, 369 → 6-AM: *m*/*z* 364 (quant), 423 → Morphine: *m*/*z* 364 (quant), 477 → Codeine: *m*/*z* 282 (quant), 395 → Deuterated IS: *m*/*z* 367 (D_3_-6-AM), 367 (D_3_-morphine), 285 (D_3_-codeine)	0.05 ng/mg hair (with a 100-mg sample).	5–1000 ng/mg	56.1 and 77.3% of drug		7 cases 20 cases 20 cases 15 cases	DIM chronic users. this appears to be the first instance of the detection of DIM and 6-acetylmorphine	[15]
DIM, 6-MAM, Morphine	Blood, Plasma, Saliva, Urine	Solvent: (J.T. Baker, Phillipsburg, NJ, USA) Sample (1 mL) + pH 6.0 acetate buffer (1 mL) + deionized water (6 mL) → Add deuterated IS (Radian Corp., Austin, TX, USA) (25 ng) + acetate buffer → Mix & centrifuge (10 min) → Filter using fritted reservoir → SPE (ZSDAUO2O, (Worldwide Monitoring, Horsham, PA, USA): conditioning (Methanol × 2, Water × 2, Acetate buffer × 2) → Sample loaded, washed (Water × 2, Acetate buffer × 2, Acetonitrile × 4) → Vacuum dry (5 min) → Elution (4 × 1 mL ethyl acetate/diethylamine 98:2) → Evaporate under N_2_ (50 °C) → Reconstitute (Acetonitrile for DIM, MBTFA (Pierce, Rockford, IL, USA) for 6-MAM & morphine, heat 60 °C for 30 min) → GC–MS Analysis	GC System: (HP 5890A → Autosampler: HP 7673A → Detector: HP 5971A MSD (SIM mode, Hewlett-Packard, Palo Alto, CA, USA) → Column: Rtx-5 capillary (15 m × 25 mm i.d., 0.10 µm film, Restek, Bellefonte, PA, USA)) → Carrier Gas: Helium (1.2 mL/min) → Injection Mode: Splitless (3 µL injection, purge time 0.50 min) → Oven Temp: 150 °C → Ramp to 200 °C → 290 °C → Detector: Mass selective (SIM mode) → Monitored Ions: DIM (*m*/*z* 327), 6-MAM (*m*/*z* 364), Morphine (*m*/*z* 364), Internal Std (*m*/*z* 367, 334) → Electron multiplier voltage: 600–800 eV above tune value → Daily tuning with perfluorotributylamine (*m*/*z* 219, 414, 502) → Data Acquisition: Selected ion monitoring (SIM) mode	1.0 ng/mL.	DIM: 1.0–250 ng/mL; 6-AM and Morphine: 1.0–500 ng/mL	Hydrolysis of DIM to 6-acetylmorphine during extraction and analysis was <5%	Within-run: 1.1–8.9% Between-run: 3.7–6.4%	DIM stable for 12–18 h in acetonitrile6-MAM & Morphine stable for 24 h post-derivatization.	corroborate DIM use and to study the pharmacological effects of DIM and its metabolites.	[34]
	Blood	Recruit 6 healthy male DIM-experienced volunteers → Ensure opioid-free status via urine testing (3 consecutive days negative) → Administer DIM hydrochloride via 3 conditions: 6 mg intranasal (IN), 12 mg IN and 6 mg intramuscular (IM), plus placebo → IN DIM mixed with lactose (total 100 mg) → Subjects inhale equal portions into each nostril using a straw → Blood samples collected pre-dose and at multiple time points post-dose (up to 24 h) → Samples frozen immediately (either plasma harvested or whole blood frozen in dry ice-acetone bath)	GC System: HP 5890A (Hewlett-Packard, Palo Alto, CA, USA) → Column: Rtx-5 capillary (15 m × 0.25 mm i.d., 0.10 µm film, Restek, Bellefonte, PA, USA)) → Carrier Gas: Helium (1.2 mL/min) → Injection Mode: Splitless (3 µL injection, purge time 0.50 min) → Oven Temp: 150 °C → Ramp to 200 °C → 290 °C → Detector: Mass selective (SIM mode) → Monitored Ions: DIM (*m*/*z* 327), 6-MAM (*m*/*z* 364), Morphine (*m*/*z* 364), Internal Std (*m*/*z* 367, 334) → Electron multiplier voltage: 600–800 eV above tune value → Daily tuning with perfluorotributylamine (*m*/*z* 219, 414, 502) → Data Acquisition: Selected ion monitoring (SIM) mode	1.0 ng/mL.	DIM: 1.0–250 ng/mL; 6-AM and Morphine: 1.0–500 ng/mL	Hydrolysis of DIM to 6-acetylmorphine during extraction and analysis was <5%	Within-run: 1.1–8.9% Between-run: 3.7–6.4%	6 healthy males		[11]
DIM, 6-MAM, Morphine	Hair	Collect hair samples from subjects → Segment hair into 1 mm (or less) sections→ Wash hair with methanol (3× vortex mixing, centrifuge at 1000× *g*, remove solvent) → Dry hair under nitrogen flow → Weigh 10 mg of hair into reaction vials → Add silylating solution (MSTFA + dithioerythritol ( + ammonium iodide, Pierce Chemical Co., Rockford, IL, USA) with internal standard (nalorphine-TMS, Sigma, St. Louis, MO, USA)) → Vortex mix for 10 s → Heat at 130 °C for 1 h → Cool sample and inject into GC–MS/MS system	GC–MS/MS system → Finnigan TSQ 700 Triple Quadrupole MS (Finnigan MAT, San Jose, CA, USA) coupled with Varian 3400 GC (Varian, Sunnyvale, CA, USA) → Column: J & W DB-5 fused silica capillary (25 m × 0.25 mm i.d., 0.25 µm film thickness, J&W Scientific, Folsom, CA, USA) → Carrier Gas: Helium (1.0 mL/min) → Injection Mode: Splitless (2 min) → Injection Port Temperature: 280 °C Oven Temperature Program → Hold at 150 °C for 2 min → Ramp 38 °C/min to 200 °C → Ramp 20 °C/min to 290 °C, hold for 7 min → Total run time: ~21 min Mass Spectrometry Conditions → Ionization Mode: Electron Impact (EI, 70 eV) → Collision Gas: Argon (0.8 mtorr) → Detection Mode: Multiple Selected Reaction Monitoring (SRM) Monitored Ions (SRM Mode) → DIM (*m*/*z* 369 → 204, 310, 327) → 6-MAM-TMS (*m*/*z* 399 → 204, 287, 340) → Morphine-2TMS (*m*/*z* 429 → 234, 287, 401) → Acetylcodeine (*m*/*z* 341 → 162, 229, 282) → Codeine-TMS (*m*/*z* 371 → 178, 234, 343)	25 pg/mg	n.a. ^¥^	n.a.	n.a.	Degradation of DIM was reduced to less than lo%	Method optimization for hair testing	[35]
DIM, 6-MAM, Morphine	Blood, Urine, Tissue	Sample Preparation → Condition SPE cartridge (ZSDAU020, Worldwide Monitoring, Horsham, PA, USA) with elution solvent (Burdick & Jackson Labs., Muskegon, MI, USA), methanol, deionized water, acetate buffer → Add liquid biological specimens (blood, plasma, saliva, urine) → Wash with deionized water, acetate buffer, acetonitrile → Aspirate to dryness → Collect analytes in elution solvent → Divide extract into two equal aliquots → Evaporate aliquots → Add acetonitrile to one set, transfer to GC–MS vials for DIM analysis → Add MBTFA (Pierce Chemical Co., Rockford, IL, USA) to second set, incubate at 60 °C for 30 min → Cool samples, transfer to GC–MS vials for 6-MAM & morphine analysis	GC System: HP 5890A Series II GC + HP 7673A autosampler + HP 5971A MSD (Hewlett-Packard, Palo Alto, CA, USA) → Column: Rtx-5 capillary (15 m × 0.25 mm i.d., 0.10 µm film) → Carrier Gas: Helium (1.2 mL/min) → Injection Mode: Splitless (3 µL injection, purge time 0.50 min) → Oven Temp: 150 °C → Ramp to 200 °C → 290 °C → Detector: Mass selective (SIM mode) → Monitored Ions: DIM (*m*/*z* 327), 6-MAM (*m*/*z* 364), Morphine (*m*/*z* 364), Internal Std (*m*/*z* 367, 334) → Electron multiplier voltage: 600–800 eV above tune value → Daily tuning with perfluorotributylamine (*m*/*z* 219, 414, 502) → Data Acquisition: Selected ion monitoring (SIM) mode (Hewlett-Packard, Palo Alto, CA, USA).	1.0 ng/mL	DIM: 2.5–100 ng/mL 6-MAM & Morphine: 10–500 ng/mL	Hydrolysis of DIM to 6-acetylmorphine during extraction and analysis was <5%,	Within-run CV: 1.1–8.9%, Between-run CV: 3.7–6.4%	DIM stable for 12–18 h in acetonitrile 6-MAM & Morphine stable for 24 h post-derivatization	Postmortem Toxicology-DIM related deaths.	[3]
DIM, 6-MAM Morphine, Normorphine	Plasma Saliva Urine, Hair	Hair Sample Preparation → Wash hair with methanol & deionized water → Dry & pulverize hair → Digest with acetate buffer (pH 4.0) + enzymatic hydrolysis (β-glucuronidase/arylsulfatase, Boehringer Mannheim, Indianapolis, IN, USA) → SPE ZSDAU020 (Worldwide Monitoring, Horsham, PA, USA): conditioning (Methanol × 2, Water × 2, Acetate buffer × 2) → Load digested sample onto SPE column → Wash (Water × 2, Acetate buffer × 2, Acetonitrile × 4) → Vacuum dry (5 min) → Elution (4 × 1 mL ethyl acetate/diethylamine 98:2) → Evaporate under N_2_ (50 °C) → Reconstitute (Acetonitrile for DIM, MBTFA (Pierce, Rockford, IL, USA) for 6-MAM & morphine, heat 60 °C for 30 min) → GC–MS Analysis Plasma, Urine and Saliva Sample Preparation → Mix sample with acetate buffer (pH 4.0, J.T. Baker, Phillipsburg, NJ, USA) + deuterated IS (25 ng) → Centrifuge (10 min) → Filter using fritted reservoir → SPE conditioning (Methanol × 2, Water × 2, Acetate buffer × 2) → Load sample onto SPE column → Wash (Water × 2, Acetate buffer × 2, Acetonitrile × 4) → Vacuum dry (5 min) → Elution (4 × 1 mL ethyl acetate/diethylamine 98:2) → Evaporate under N_2_ (50 °C) → Reconstitute (Acetonitrile for DIM, MBTFA for 6-MAM & morphine, heat 60 °C for 30 min) → GC–MS Analysis	GC–MS Analysis Method → GC System: HP 5890A + HP 7673A autosampler + HP 5970B MSD (Hewlett-Packard, Palo Alto, CA, USA) → Column: HP-1 capillary (12 m × 0.20 mm i.d., 0.33 µm film, Agilent/HP, Palo Alto, CA, USA) → Carrier Gas: Helium (1 mL/min) → Injection Mode: Splitless → Oven Temp: 120 °C → Ramp to 220 °C → 250 °C → 260 °C → Detector: Mass selective (SIM mode) → Monitored Ions: DIM (*m*/*z* 327), 6-MAM (*m*/*z* 364), Morphine (*m*/*z* 364), Internal Std (*m*/*z* 367, 334) → Electron multiplier voltage: 600–800 eV above tune value → Data Acquisition: Selected ion monitoring (SIM) mode (Hewlett-Packard, Palo Alto, CA, USA)	Hair Samples: LOD: 0.1 Plasma, Sérum, Urine samples: LOD: 1	Hair Samples: 0.1–10 ng/mg and 10–100 ng/mg Plasma, Sérum, Urine samples: 2.5–500 ng/mL	>90%	Hair Samples Within-run CV: 3–26.5% Between-run CV: 5–15% Plasma, Serum, Urine samples: Within-run CV: 4–12% Between-run CV: 6–14%	Hydrolysis of DIM to 6-acetylmorphine during extraction and analysis was 10%,	Forensic Toxicology– Hair for long-term DIM use. P.S.U for recent DIM used.	[36]
DIM 6-MAM	Sweat	Apply sweat patch (PharmChem Inc., Menlo Park, CA, USA) to subject’s skin → Clean skin with 70% isopropyl alcohol ((Fisher Scientific, Fair Lawn, NJ, USA) → Secure patch firmly → Wear patch for designated period (up to several days) → Remove patch carefully, avoiding contamination → Freeze patch at −30 °C until analysis → Extract sweat from absorbent pad using acetate buffer + Triton X-100 → Centrifuge (10 min at 2000 rpm) → Filter extract through SPE column ((Clean Screen® DAU, 200 mg, 10 mL; United Chemical Technologies, Bristol, PA, USA)) → Wash with methanol (J.T. Baker, Phillipsburg, NJ, USA), deionized water, acetate buffer → Elute analytes (ethyl acetate-diethylamine for DIM metabolites) → Evaporate under nitrogen → Reconstitute (N,O-bis(trimethylsilyl)trifluoroacetamide (BSTFA, Pierce Chemical Co., Rockford, IL, USA) for derivatization of 6-MAM & morphine, heat at 60 °C for 30 min) → GC–MS Analysis	GC System: HP 5890A (Hewlett-Packard, Avondale, PA, USA))→ Column: HP-1 capillary (12 m × 0.20 mm i.d., 0.33 µm film, Hewlett-Packard, Avondale, PA, USA) for cocaine, Rtx-5 capillary (15 m × 0.25 mm i.d., 0.10 µm film) for DIM → Carrier Gas: Helium (1 mL/min for cocaine, 1.2 mL/min for DIM) → Injection Mode: Splitless → Oven Temp: Cocaine (120 °C → Ramp to 220 °C → 250 °C → 260 °C), DIM (150 °C → Ramp to 200 °C → 290 °C) → Detector: Mass selective (SIM mode) → Monitored Ions: DIM (*m*/*z* 327), 6-MAM (*m*/*z* 364), Morphine (*m*/*z* 364), Internal Std (*m*/*z* 367, 334) → Electron multiplier voltage: 600–800 eV above tune value → Daily tuning with perfluorotributylamine (*m*/*z* 219, 414, 502) → Data Acquisition: Selected ion monitoring (SIM) mode.	1.0 ng per patch	2.5 to 50 ng per patch	75–95%	less than 10%,	Not specified	Monitoring DIM and cocaine use in clinical studies	[16]
DIM, 6-MAM, Morphine	Blood	Prepare DIM base solution in acetonitrile (Burdick & Jackson, Muskegon, MI, USA) → Apply precise dose to nichrome wire coil → Air-dry overnight → Verify weight within ±20% of target → Load coil into smoking device → Activate heating element (≤200 °C) to volatilize DIM → Inhale single puff, hold for 15 s → Collect blood samples at specific time intervals → Store blood at −30 °C until analysis → Extract DIM and metabolites using SPE (ZSDAU020, (Worldwide Monitoring, Horsham, PA, USA)) → Elute analytes with ethyl acetate-2% diethylamine → Evaporate under nitrogen → Reconstitute (Acetonitrile for DIM, MBTFA (Pierce Chemical Co., Rockford, IL, USA) for 6-MAM & morphine, heat at 60 °C for 30 min) → GC–MS Analysis	GC System: HP 5890A (Hewlett–Packard, Avondale, PA, USA) → Column: Rtx-5 capillary (15 m × 25 mm i.d., 0.10 µm film, Restek, Bellefonte, PA, USA) → Carrier Gas: Helium (1.2 mL/min) → Injection Mode: Splitless (3 µL injection, purge time 0.50 min) → Oven Temp: 150 °C → Ramp to 200 °C → 290 °C → Detector: Mass selective (SIM mode) → Monitored Ions: DIM (*m*/*z* 327), 6-MAM (*m*/*z* 364), Morphine (*m*/*z* 364), Internal Std (*m*/*z* 367, 334) → Electron multiplier voltage: 600–800 eV above tune value → Daily tuning with perfluorotributylamine (*m*/*z* 219, 414, 502) → Data Acquisition: Selected ion monitoring (SIM) mode	LOD: 1 ng/mL for DIM and metabolites.	LOD: 1.0 ng/mL DIM: 1.0–250 ng/mL, 6-MAM & Morphine: 1.0–500 ng/mL	Hydrolysis of DIM to 6-acetylmorphine during extraction and analysis was <5%	Within-run: 1.1–8.9% Between-run: 3.7–6.4%	Recovery studies indicated that smoking device delivered approximately 89% of DIM	Comparison of smoked vs. intravenous DIM pharmacokinetics in experienced DIM users	[12]
DIM, 6-MAM Morphine	Blood Urine Vitreous humor	Biological fluid (1 mL) + Internal standard (Diethylnalorphine, 100 µL, 10 mg/L) → Add sodium carbonate-sodium bicarbonate buffer (~10 mg) + sodium chloride (~250 mg) → Liquid–liquid extraction with ethyl acetate–chloroform–hexane (7:2:1, 20 mL) (Fisher Scientific, Ottawa, Canada) → Shake (10 min) + Centrifuge (5 min, 2000 rpm) → Transfer organic phase (10 mL) to round-bottom flask → Evaporate under nitrogen (70 °C sand bath) → Add catalyst solution (4-dimethylaminopyridine, pyridine, chloroform, Aldrich, Milwaukee, WI, USA)) → Propionylation with propionic anhydride (50 µL, vortex 10 sec, react 30 min at room temp, (Aldrich, Milwaukee, WI, USA) → Quench reaction with methanol (50 µL, vortex 15 s, stand 1 min) → Eliminate propionic acid via hexane azeotrope (2 × 500 µL, nitrogen stream, 80 °C sand bath) → Reconstitute in chloroform-0.1% pyridine (250 µL) → GC–MS Analysis	GC System: Varian 3400 gas chromatograph + Saturn II ion trap detector ((Varian, Mississauga, ON, Canada) + Varian 8100 Autosampler (Varian, Mississauga, ON, Canada) → Column: DB-1 fused silica capillary (30 m × 0.32 mm i.d., 0.25 µm film thickness, Chromatographic Specialties, Brockville, ON, Canada) → Carrier Gas: Helium (99.999%, 16 psi, 3.6 mL/min) → Injection Mode: Programmable temperature injector (SPI) → Oven Temp: 40 °C (3 min) → Ramp 20 °C/min to 200 °C (1 min) → Ramp 5 °C/min to 260 °C → MS Transfer Line Temp: 300 °C → Monitored Ions: DIM (*m*/*z* 310, 327, 369), 6-AM (*m*/*z* 327, 382, 383), Morphine (*m*/*z* 324, 341, 396), Internal Std (*m*/*z* 322, 338, 367) → Electron impact mode (23 Pa filament emission current) → Data Acquisition: Full-scan (250–405 amu), 8 scans/s → Daily tuning with perfluorotributylamine (50–650 amu range)	2 (2–5) 2 (50) 2 (100)	10–50 50–250 100–500	>95% ~80% ~80%	5–12%. (4–15%)	DIM (Stable for 12–18 h in chloroform) → 6-MAM (Stable for 24 h post-derivatization) → Morphine (Stable for 24 h post-derivatization)	Forensic toxicology—postmortem DIM detection	[37]
DIM 6-MAM Morphine M3G M6G	Serum	Collect blood samples at specific time intervals → Store plasma at −30 °C until analysis → Thaw plasma before extraction → Add ammonium bicarbonate buffer (500 µL, 0.1 M, pH 9.2, Merck, Darmstadt, Germany) to 250 µL plasma → Solid-phase extraction (SPE) using C8, 50 mg columns (Varian, Harbor City, CA, USA) → Reconstitute residue with 100 µL methanol → Inject 80 µL into system.	HPLC Analysis Method → HPLC System: Hewlett-Packard 1050 series LC pump (Waldbronn, Germany) → Column: Nucleosil 100 5 C18 AB reversed-phase (250 mm × 4 mm, 5 µm, (Macherey-Nagel, Düren, Germany) → Mobile Phase: Triethylammonium phosphate (TEAP) buffer, pH 7.0; prepared from 1 M stock and diluted 1:40 with distilled water → Detection: Fluorescence (Excitation: 220 nm, Emission: 340 nm) → Injection Volume: 80 µL → Analytes: Morphine-3-glucuronide, Morphine-6-glucuronide → Calibration Range: Morphine-3-glucuronide (5–500 ng/mL), Morphine & Morphine-6-glucuronide (15–500 ng/mL) GC–MS Analysis Method: [34]	n.d. n.d. 3 10 10	n.a. n.a. 15–500 5–500 15–500	n.a. n.a. 71 71 62	n.a. n.a. 3–6 4–6 4–7	DIM, 6-MAM (GC–MS), MOR, M3G, M6G (HPLC)	Postmortem toxicology application for DIM-related death investigations	[30]
DIM 6-MAM Morphine M3G M6G Normorphine C6G Codeine	Plasma	Plasma sample (1.5 mL) + Internal standard (Ethylmorphine, 150 µL, 0.1% aqueous solution, Fluka, Buchs, Switzerland) → Add phosphate buffer (4.5 mL, 0.01 M, pH 2.1, Merck, Darmstadt, Germany), vortex for 10 sec → Load onto Nucleosil C18 ODS-2 SPE column (Macherey-Nagel, Düren, Germany) → 0% acetonitrile in phosphate buffer (pH 2.1) → Bi-distilled water (0.5 mL) → Acetonitrile-phosphate buffer (0.35 mL, 40%) → Elute analytes with acetonitrile-phosphate buffer (2 × 0.6 mL, 40%) → Evaporate eluate to dryness → Reconstitute in bi-distilled water (100 µL) → Inject 2 µL into HPLC system	HPLC System: Hewlett-Packard 1090 Series II (Hewlett-Packard, Waldbronn, Germany) → Column: Spherisorb C18 ODS-2 (125 × 2 mm, 3 µm, Stagroma, Wallisellen, Switzerland) + Guard column (20 × 2 mm, Stagroma, Wallisellen, Switzerland) → Mobile Phase: Multi-step gradient (all solvents HPLC-grade from Merck, Basel, Switzerland and Fluka, Buchs, Switzerland) (Water–phosphoric acid–hexylamine + Acetonitrile–water–phosphoric acid–hexylamine) → Flow Rate: 0.25 mL/min → Column Temperature: 30 °C → Detection: Diode-array (DAD, Hewlett-Packard, Waldbronn, Germany) at 210 nm → Online UV spectra recorded (192–350 nm) for peak purity and assignment	(25) (25) (25) (25) (25) (25) (25)	25–5000	88 94 98 91 90 88 100 91	3.1–4.8 2.5–4.4 1.6–4 2.5–5 2.2–4.7 2–4.6 3.2–4.8 2.2–4.4		Forensic toxicology—monitoring DIM and metabolite concentrations in plasma samples	[31]
DIM 6-MAM Morphine M3G M6G Codeine	Serum	Serum sample (1 mL) + Internal standard (Nalorphine, 200 µL, 1 µg/mL aqueous solution, Sigma, St. Louis, MO, USA) → SPE using ethyl SPE columns (1 mL, 100 mg sorbent, J.T. Baker, Phillipsburg, NJ, USA) → Condition SPE column (Methanol × 2, Water × 1, Ammonium hydrogen carbonate buffer (pH 9.3, 2 mL), Merck, Darmstadt, Germany) → Load serum sample onto SPE column → Wash (1 mL Ammonium Hydrogen Carbonate buffer → Elute analytes with 1 mL methanol → Evaporate eluate under nitrogen at room temperature → Reconstitute in mobile phase (100 µL methanol–acetonitrile–water–formic acid) → Inject 5 µL into LC–MS system	LC System: Applied Biosystems 140B dual-syringe solvent-delivery pump (Applied Biosystems, Foster City, CA, USA)→ Column: Supelcosil LC–Si (25 cm × 2.1 mm, 5 µm particle size, Supelco Inc., Bellefonte, PA, USA) → Mobile Phase: Methanol-acetonitrile-water-formic acid (59.8:5.2:34.65:0.35, *v*/*v*/*v*/*v*) → Flow Rate: 230 µL/min (split to 46 µL/min before MS inlet) → Ionization Source: Atmospheric pressure chemical ionization (API I single quadrupole mass spectrometer (Thornhill, Ontario, Canada) equipped with an atmospheric pressure ionspray (API) interface) → Capillary Tip Voltage: 5000 V → Orifice Voltage: 50 V (M3G, M6G), 70 V (Nalorphine, Morphine, DIM, Codeine) → Vacuum Pressure: 1.8 × 10^−5^ torr → Source Temperature: 60 °C → Carrier Gas: Nitrogen → Detection Mode: Selected-ion monitoring (SIM) → Monitored Ions: DIM (*m*/*z* 370), 6-MAM (*m*/*z* 328), Morphine (*m*/*z* 286), M3G (*m*/*z* 462), M6G (*m*/*z* 462), Codeine (*m*/*z* 300), Internal Std (*m*/*z* 312)	(0.5) (4) (4) (1) (4) (4)	0.5–10 4–10 4–10 1–10 4–10 4–10	72–75 99–100 99 77–80 44–46 99–100	3.6–5.3 1–1.5 1.1–2.2 3.6–6.7 4–6.2 1–1.6		DIM-related pharmacokinetic studies	[38]
DIM, 6-MAM, Morphine	Sweat	Apply PharmChek sweat patch (PharmChem, Menlo Park, CA, USA) to upper back → Clean skin with 70% isopropanol before application → Wear patch for 24 h before removal → Store absorbent pad at −20 °C in sealed plastic tubes → Extract target drugs using 5 mL acetonitrile (Merck, Darmstadt, Germany) with internal standards (DIM-d_9_, 6-MAM-d_3_, morphine-d_3_) → Agitate for 30 min at 200 rpm → Divide extract into two portions (2 mL for DIM, remainder for other compounds) → Evaporate acetonitrile to dryness → Reconstitute DIM fraction in acetonitrile → Derivatise remaining fraction with BSTFA + TMCS (60 °C, 20 min, (Interchim, Montluçon, France) → Inject into GC–MS system	GC System → Hewlett-Packard 5890 GC coupled with HP 5989B MSD (Hewlett-Packard, Les Ulis, France) → Column: HP-5MS fused-silica capillary (30 m × 0.25 mm i.d., 0.25 µm film thickness, Hewlett-Packard, Avondale, PA, USA) → Carrier Gas: Helium (1.0 mL/min) → Injection Mode: Splitless (1.5 µL sample) → Injection Port Temperature: 260 °C Oven Temperature Program → Hold at 60 °C for 1 min → Ramp 30 °C/min to 290 °C, hold for 6 min → Total run time: ~12 min Mass Spectrometry Conditions → Ion Source Temperature: 230 °C → Ionization Mode: Electron Impact (EI, 70 eV) → Detection Mode: Selected-ion monitoring (SIM) Monitored Ions (SIM Mode) → DIM (*m*/*z* 310, 327, 369), HER-d_9_ (*m*/*z* 378) → 6-MAM-TMS (*m*/*z* 340, 399), 6-MAM-d_3_-TMS (*m*/*z* 402) → Morphine-TMS (*m*/*z* 401, 414, 429), Morphine-d_3_-TMS (*m*/*z* 432)	0.5 1.0 1.0	2.1–96.3 (DIM), 0–24.6 (6–MAM), 0–11.2 (morphine)	71 76 71	<13%	hydrolysis of DIM and 6-acetylmorphine was, 4% over that period of one month.	Forensic toxicology drug monitoring using sweat patches	[17]
DIM 6-MAM Morphine M3G M6G	Plasma Urine	Recruit 2 opioid-dependent patients enrolled in a DIM-assisted treatment program → Administer diacetylmorphine (DAM) via three routes: → Intravenous (IV): 200 mg bolus → Oral: Capsules (2 × 400 mg) and controlled-release tablets (460 mg + 690 mg) → Rectal: Suppositories (2 × 400 mg) → Collect blood samples at multiple time points (pre-dose to 720 min post-dose) via peripheral venous catheter → Collect urine samples for metabolite profiling → Store samples appropriately for chromatographic analysis. Analytical methods (HPLC–DAD for plasma and GC–MS for urine) were performed according to the validated procedures previously described by Inturrisi et al [14].	Analytes → DIM, 6- 6-MAM, morphine, M3G, and M6G→ Instrumentation → Plasma analysis: HPLC, DAD, → Urine analysis: GC–MS Detection → Quantification of DIM and metabolites in plasma and urine AS previously described by Inturrisi et al [14].	n.a.	n.a.	n.a.	n.a.		Pharmacodynamics and pharmacokinetics	[39]
DIM, 6-MAM, Morphine, M3G, M6G Codeine C6G	Blood, Plasma	Collect arterial and venous blood samples at specific time intervals → Store plasma at −20 °C until analysis → Thaw plasma before extraction → Add fluoride and heparin to vacuum tubes for stabilization →Dilute 1 mL plasma with 2 mL 0.01 M ammonium carbonate buffer (pH 9.3; Merck, Darmstadt, Germany) → Solid-phase extraction (SPE, Bond Elut C18 (200 mg, (Varian, Harbor City, CA, USA) using reversed-phase cartridges → Wash with 0.01 M ammonium carbonate buffer (pH 9.3) → Elute analytes with 0.5 mL methanol—0.5 M acetic acid (9:1 *v*/*v*; Merck, Darmstadt, Germany) → Evaporate under nitrogen → Reconstitute in mobile phase (100 µL methanol–acetonitrile–water–formic acid) → Inject into LC–MS system	LC System →Instrument: Merck–Hitachi Model 6200 gradient pump (Merck–Hitachi, Darmstadt, Germany) → Injection Valve: Rheodyne Type 7125 (20 µL loop) (Rheodyne, Berkeley, CA, USA) → Column: Superspher RP 18 (125 × 3 mm I.D., 4 µm particle size, Merck, Darmstadt, Germany) → Mobile Phase: Acetonitrile–50 mM ammonium formate buffer, pH 3.0 (5:95, *v*/*v*) → Flow Rate Program: 0.6 mL/min for 4 min → Increase to 1.1 mL/min over 3 min → Hold at 1.1 mL/min for 10 min ESI (+)–MS Analysis →Instrument: Finnigan MAT SSQ 7000 single quadrupole (Finnigan MAT, San Jose, CA, USA) → Ionization Source: APCI → Sheath Gas: Nitrogen (70 psi) → Auxiliary Gas: Nitrogen (20 mL/min) → Heated Vaporizer Temperature: 450 °C → Heated Capillary Temperature: 180 °C → Corona Current: 5 µA → Mass Spectra Range: 100–500 u → Octapole Offset: 10 V (positive-ion mode) → Detection Mode: Selected-ion monitoring (SIM) → SIM Time Windows & Monitored Ions → 0–5 min → *m*/*z* 286, 289, 462, 465 5–11 min → *m*/*z* 300, 303, 306, 476, 479 11–17 min → *m*/*z* 328, 334 Scan Time → 0.5 s	0.5–100	n.a.	n.a.	n.a.	Linear up to high doses	Arterial and venous pharmacokinetics of DIM metabolism in addicts	[8]
DIM 6-MAM Morphine	Urine	Collect urine samples from suspected DIM users → Directly inject into column-switching HPLC system → Use strong cation-exchange (SCX, Capcell Pak MF SCX 2 × 10 mm, 5 µm (Shiseido, Tokyo, Japan) column for on-line SPE → Remove endogenous urinary constituents during trapping phase → Enrich DIM and metabolites on SCX column (Capcell Pak MF SCX (2.0 × 10 mm, 5 µm; Shiseido, Japan) → Transfer enriched analytes to analytical column via backflush mode → Elute with ammonium acetate–acetonitrile mobile phase (Merck, Darmstadt, Germany) → Introduce entire eluent into electrospray ionization mass spectrometry (ESI–MS) without splitting	Instrumentation → Column-switching HPLC (Shiseido, Tokyo, Japan) coupled with ESI–MS (Platform quadrupole, Micromass, Manchester, UK) → Columns: Trapping column: CAPCELL PAK MF SCX (2.0 mm × 10 mm, 5 µm, Shiseido, Japan)) → Analytical column: CAPCELL PAK SCX (1.5 mm × 150 mm, 5 µm, Shiseido, Japan) Mobile Phase → Main separation: 10 mM ammonium acetate (pH 6.0)–acetonitrile (30:70, *v*/*v*) Flow Rates → Trapping: 200 µL/min → Main separation: 120 µL/min ESI–MS Conditions → Ion source temperature: 70 °C → Capillary voltage: +4.5 kV → Cone voltage: 50 V → Detection Mode → Selected Ion Monitoring (SIM) → Target Ions → DAM: *m*/z 370 → 6-MAM: *m*/*z* 328 → Morphine: *m*/*z* 286	0.1 0.5 3	1–100 1–100 10–1000	99 101 104	3.7 4.1 3.4		opiates in users’ urine samples	[18]
DIM, 6-MAM Morphine Normorphine	Urine	Urine sample (100 µL) + Internal standard (Cerilliant, Austin, TX, USA)) (10 µL working solution) → Vortex-mix briefly → Centrifuge (5 min at 510 g) to remove large particles → Inject 10 µL supernatant directly into LC–MS/MS system	LC System: Surveyor HPLC system, LCQ Deca XP ion trap with orthogonal APCI source (positive ion mode) (ThermoFinnigan, San Jose, CA, USA) → Column: Synergi Polar RP (150 × 2.0 mm, 4 µm, Phenomenex, Torrance, CA, USA) + guard column (4 × 2 mm) (Phenomenex, Torrance, CA, USA) → Mobile Phase: Gradient elution (A: 10 mM ammonium formate in water, 0.001% formic acid, pH 4.5; B: Acetonitrile, Sigma, St. Louis, MO, USA) → Flow Rate: 300 µL/min → Gradient Program: 0–13 min (5% B → 26%B ) → 13–22 min (26% B → 90% B) → 22–24 min (90% B) → 24–27 min (90% B → 5% B) → 27–35 min (5% B) → Column Oven Temp: 25 °C → Auto-injector Tray Temp: 4 °C → Detection Mode: Selected reaction monitoring (SRM) → Ionization Source: Atmospheric pressure chemical ionization (APCI) → Corona Discharge Needle Voltage: 4.7 kV → Vaporizer Temp: 450 °C → Sheath Gas: Nitrogen (70 psi) → Transfer Capillary Temp: 220 °C → Electron Multiplier Voltage: 850 eV → Monitored Ions: DIM (*m*/*z* 370 → 310, 328, 268), 6-MAM (*m*/*z* 328 → 268, 211, 193), Morphine (*m*/*z* 286 → 268, 229, 211), Internal Std (*m*/*z* varies per analyte)	Between 10–100	10–10,000	>80	3.1–16	Analyte stability was accepted with a recovery greater than 80%.	Applicable in drug abuse screening and methadone treatment monitoring	[9]
DIM, 6-MAM, Morphine, M3G, M6G	Plasma	Collect arterial and venous blood samples at specific time intervals → Store plasma at −20 °C until analysis → Thaw plasma before extraction → Add fluoride and heparin to vacuum tubes for stabilization → Solid-phase extraction (SPE, Bond Elut C18 (200 mg, Varian, Harbor City, CA, USA) using reversed-phase cartridges → Condition cartridges with methanol, water, and ammonium formate buffer (pH 3.0) → Wash with ammonium formate buffer (pH 3), methanol and acetonitrile → Elute analytes with methanol-acetonitrile gradient → Evaporate under nitrogen → Reconstitute in mobile phase (100 µL methanol–acetonitrile–water–formic acid) → Inject into LC–MS system	LC System →Instrument: Merck–Hitachi Model L2000 gradient pump (Merck, Darmstadt, Germany) → Injection Valve: Rheodyne Type 8125 (20 µL loop, Cotati, CA, USA) → Column: Superspher RP 18 (125 × 3 mm i.d., 4 µm particle size, Merck, Darmstadt, Germany) → Mobile Phase: Acetonitrile–50 mM ammonium formate buffer, pH 3.0 (5:95, *v*/*v*) → Flow Rate Program: 0.6 mL/min for 4 min → Increase to 1.1 mL/min over 3 min → Hold at 1.1 mL/min for 10 min APCI–MS Analysis →Instrument: Finnigan MAT SSQ 7000 single quadrupole (Finnigan MAT, San Jose, CA, USA) → Ionization Source: Atmospheric Pressure Chemical Ionization (APCI) → Sheath Gas: Nitrogen (70 psi) → Auxiliary Gas: Nitrogen (20 mL/min) → Heated Vaporizer Temperature: 400 °C → Heated Capillary Temperature: 170 °C → Corona Current: 5 µA → Mass Spectra Range: 100–500 u → Octapole Offset: 10 V (positive-ion mode) → Detection Mode: Selected-ion monitoring (SIM) → SIM Time Windows & Monitored Ions → 0–5 min → *m*/*z* 286, 289, 462, 465 5–11 min → *m*/*z* 300, 303, 306, 476, 479 11–17 min → *m*/*z* 328, 334 Scan Time → 0.5 s	LLQ: 1 nmol/L (DIM, 6-MAM), 10 nmol/L (Morphine, M3G, M6G)	Linear pharmacokinetics observed for intramuscular and oral DIM administration	n.a.	n.a.	n.a.	Comparison of intramuscular vs. oral DIM pharmacokinetics in opioid-dependent patients receiving high-dose maintenance therapy	[40]
DIM 6-MAM Morphine	Hair	Collect hair samples from subjects in DIM-maintenance program and opiate-associated fatalities → Segment hair into 1 cm sections for analysis → all chemical obtained from (Merck, Darmstadt, Germany): Wash hair samples with dichloromethane (2×, 5 mL, 2 min each) → Dry and cut into small pieces (~30 mg) → Incubate in methanol (1 mL, 45 °C, 16 h) with internal standards (DIM-d_3_, 6-MAM-d_3_, morphine-d_3_) → Evaporate methanol extract to dryness → Derivatise residue with N-Methyl-N-(trimethylsilyl) trifluoroacetamide (MSTFA), pyridine and iso-octane at 90 °C for 15 min, Düren, Germany) → Inject into GC–MS system	GC System → Agilent 6890 Plus GC coupled with Agilent 5973N MSD (Chromtech, Idstein, Germany) → Column: HP-5MS fused-silica capillary (30 m × 0.25 mm i.d., 0.25 µm film thickness, Agilent Technologies, Palo Alto, CA, USA)) → Carrier Gas: Helium (1.0 mL/min) → Injection Mode: Splitless (1 µL sample) → Injection Port Temperature: 280 °C Oven Temperature Program → Hold at 180 °C for 1 min → Ramp 15 °C/min to 190 °C, hold for 10 min → Ramp 5 °C/min to 250 °C → Increase at 30 °C/min to 290 °C, hold for 2 min → Total run time: 21.28 min Mass Spectrometry Conditions → Ion Source Temperature: 230 °C → Quadrupole Temperature: 150 °C → Transfer Line Temperature: 290 °C → Ionization Mode: Electron Impact (EI, 70 eV) → Detection Mode: Selected-ion monitoring (SIM) Monitored Ions (SIM Mode) → DIM (*m*/*z* 369, 327, 268), HER-d9 (*m*/*z* 378, 334, 272) → 6-MAM-TMS (*m*/*z* 399, 340, 287), 6-MAM-d3-TMS (*m*/*z* 402, 343, 290) → Morphine-TMS (*m*/*z* 429, 414, 236), Morphine-d3-TMS (*m*/*z* 432, 417, 239) → Codeine-TMS (*m*/*z* 371, 234, 196), Codeine-d3-TMS (*m*/*z* 374, 237, 199) → Acetylcodeine (*m*/*z* 229, 341, 282), Acetylcodeine-d3 (*m*/*z* 232, 344, 285)	0.04 (0.21) 0.02 (0.15) 0.03 (0.11) 0.02 (0.04) 0.02 (0.13)	0.5–25 0.5–25 0.5–25 0.5–25 0.5–25	79–95 91–96 92–95 80–85 62–86	<15%		Forensic toxicology—hair analysis for long-term DIM use verification	[27]
DIM, 6-MAM Morphine, M3G M6G	Plasma	Collect human plasma sample(0.25 mL) → Add internal standards: DIM-d_6_, Morphine-d_3_, Morphine-3-glucuronide-d_3_, Methadone-d_9_ (Cerilliant, Round Rock, TX, USA) → Acidify with 0.15 N HCl (VWR International, Leuven, Belgium) and add sodium fluoride (Merck, Darmstadt, Germany→ Perform SPE: Use mixed-mode sorbent columns (MCX Oasis, Waters Corporation, Milford, MA, USA) to extract analytes → all solvent were obtained from Merck, Darmstadt, Germany): Condition SPE column with methanol and aqueous buffer (Merck, Darmstadt, Germany) → Load plasma onto SPE column → Wash (Aqueous buffer × 1, Methanol × 1) → Elute analytes with 0.5% ammonium acetate in methanol (*v*/*v* 1:20) → Evaporate eluate under nitrogen at room temperature → Reconstitute in mobile phase (e.g., 100 µL methanol–acetonitrile–water–formic acid) → Inject defined volume into LC–MS/MS system	LC-MS/MS Analysis Method → LC System → Instrument: Agilent 1290 Infinity II LC (Agilent Technologies, Santa Clara, CA, USA) → Column: Zorbax Eclipse Plus C18 (2.1 × 100 mm, 1.8 µm particle size, Agilent Technologies, Santa Clara, CA, USA)→ Column: Zorbax Bonus-RP (150 × 4.6 mm, 5 µm particle size, Agilent Technologies, Palo Alto, CA, USA) →Temperature: 40 °C → Mobile Phase: Gradient elution using 5 mM ammonium formate buffer (pH 4.0, Fluka, Buchs, Switzerland) and acetonitrile (Merck, Darmstadt, Germany) → Flow Rate: 1.0 mL/min → Run Time: 15 min Mass Spectrometry Conditions → Ionization Source: Electrospray ionization (Turbo Ionspray, ESI) in positive-ion mode → Mass Spectrometer: API 3000 Triple Quadrupole (PE Sciex, Concord, ON, Canada) → Detection Mode: Multiple Reaction Monitoring (MRM) → Monitored Analytes: DIM, methadone and their metabolites (including 6-MAM, morphine, morphine-3-glucuronide, morphine-6-glucuronide), as well as cocaine, acetylcodeine and their metabolites	5	5–500	86–96 86–94 83–101 79–80 79–81	2.4–11 4.3–9.5 2.9–11 2.8–7.4 6.4–10	Short term, long-term, freeze and thaw were less than 10%	Validated method for forensic toxicology and clinical drug monitoring	[32]
DIM 6-MAM Morphine	Serum	Collect serum sample → Dilute 1:10 with micellar mobile phase (0.1 M sodium dodecyl sulfate + 4% (*v*/*v*) 1-butanol in 0.01 M phosphate buffer, pH 7.0) → Inject diluted serum (100 µL) directly into chromatographic system without extraction or derivatization.	HPLC System → Konik 500 HPLC system (Konik Instruments, Barcelona, Spain) → Column: C18 reversed-phase (250 × 4.0 mm, 5 µm particle size, Scharlab, Barcelona, Spain)) → Mobile Phase: 0.1 M sodium dodecyl sulfate-4% butanol (pH 7.0) → Flow Rate: 1.0 mL/min → Run Time: <18 min Detection → UV detection at 230 nm → Retention times: DIM (~15 min), 6-MAM (~13 min), Morphine (~11 min), Benzoylecgonine (~7 min).	23 (36) 15 (28) 11 (23)	0.02–10	90–98 98–101 99–100	1.1–2.4 0.7–1.9 0.4–1.6		forensic toxicology and clinical drug monitoring	[41]
DIM, 6-MAM, Morphine, M3G, M6G		Collect plasma samples from DIM-dependent patients → Add sodium fluoride and potassium oxalate immediately to prevent hydrolysis → Centrifuge and freeze plasma at −70 °C → Pre-treat plasma samples via SPE → Use Oasis MCX sorbent columns (Waters Corporation, Milford, MA, USA) → Apply deuterated internal standards for DAM, 6-AM, morphine and glucuronides → Follow validated procedure based on [32] → Inject extracts into HPLC–MS/MS system	[32]	5 ng/mL	[32]	[32]	5–11	n.a.		[19]
DIM, 6-MAM, Morphine M3G, M6G	Plasma	Collect arterial and venous blood samples at specific time intervals → Store plasma at −30 °C until analysis → Thaw plasma before extraction → Add sodium fluoride and potassium oxalate to tubes for stabilization → Centrifuge (2000 g, 5 min) to separate plasma → Solid-phase extraction (SPE) using reversed-phase cartridges → Wash with ammonium formate buffer (pH 3), methanol and acetonitrile → Elute analytes with methanol-acetonitrile gradient → Evaporate under nitrogen → Reconstitute in mobile phase (100 µL methanol–acetonitrile–water–formic acid) → Inject into LC–MS/MS system	[32]	LOD: 5 ng/mL for all analytes	Calibration curves linear over 0.1–50 ng/mL (r > 0.999)	[32]	[32]	Intra-day precision CV < 10%, Inter-day precision CV < 15%	Comparison of intravenous vs. inhalation DIM pharmacokinetics and pharmacodynamics in opioid-dependent patients receiving high-dose maintenance therapy	[20]
DIM, 6-MAM Morphine Codeine 6-Acetylcodeine	Sweat	Apply PharmChek sweat patch (PharmChek™, PharmChem Inc., Menlo Park, CA, USA to upper back → Clean skin with 70% isopropanol before application → Wear patch for 7 days before removal → Store absorbent pad at −20 °C in sealed plastic tubes → Elute drugs from sweat patch using 6 mL sodium acetate buffer (pH 4.0) → Perform solid-phase extraction (SPE) using Clean Screen^®^ ZSDAU020 columns (United Chemical Technologies, Bristol, PA, USA) → Wash with distilled water, acetic acid and methanol → Elute analytes with methylene chloride:2-propanol:ammonium hydroxide (78/20/2, *v*/*v*/*v*) → Evaporate eluate under nitrogen at 35 °C → Derivatise with BSTFA + TMCS (60 °C, 20 min, (Supelco, Bellefonte, PA, USA) → Inject into GC–MS system	GC System → Agilent 6890 GC coupled with Agilent 5973 MSD (Agilent Technologies, Santa Clara, CA, USA)→ Column: HP-5MS fused-silica capillary (30 m × 0.32 mm i.d., 0.25 µm film thickness, Agilent Technologies, USA) → Carrier Gas: Helium (1.1 mL/min) → Injection Mode: Splitless (2 µL sample) → Injection Port Temperature: 200 °C Oven Temperature Program → Hold at 100 °C for 0.5 min → Ramp 25 °C/min to 245 °C → Ramp 2 °C/min to 255 °C → Increase at 30 °C/min to 300 °C, hold for 0.7 min → Total run time: 13.5 min Mass Spectrometry Conditions → Ion Source Temperature: 230 °C → Quadrupole Temperature: 150 °C → Ionization Mode: Electron Impact (EI, 70 eV) → Detection Mode: Selected-ion monitoring (SIM) Monitored Ions (SIM Mode) → DIM (*m*/*z* 369, 327, 310), HER-d_9_ (*m*/*z* 378) → 6-MAM-TMS (*m*/*z* 399, 340, 287), 6-MAM-d_3_-TMS (*m*/*z* 402, 343, 290) → Morphine-TMS (*m*/*z* 429, 414, 236), Morphine-d_3_-TMS (*m*/*z* 432, 417, 239) → Cocaine (*m*/*z* 182, 303, 272), Cocaine-d_3_ (*m*/*z* 185, 306, 275) → Benzoylecgonine (*m*/*z* 240, 361, 346), Benzoylecgonine-d_3_ (*m*/*z* 243, 364, 349) → Methadone (*m*/*z* 294, 115, 165)	2.5 ng/patch	5–1000 (DIM), 10–1000 (methadone)	69–78 97–101 102–107 108–112 104–108	0.8 to 6.5%	DIM hydrolysis during specimen processing was <11%	forensic toxicology and drug monitoring using sweat patches	[42]
DIM, 6-MAM Morphine M3G M6G	Blood–Brain	Collect whole blood and brain tissue samples from mice → Immediately add ice-cold acidic buffer containing sodium fluoride (4 mg/mL, Sigma-Aldrich, USA) → Homogenize brain tissue in ammonium formate buffer (pH 3.1) → Perform protein precipitation using ice-cold acetonitrile/methanol (85:15) → Centrifuge at 4500 rpm (3900× *g*) at 4 °C for 10 min (Eppendorf 5810R, Hamburg, Germany) → Evaporate supernatant to dryness at 40 °C under nitrogen (TurboVap LV, Zymark/Caliper, Hopkinton, MA, USA) → Reconstitute residue in mobile phase (3% acetonitrile/97% ammonium formate buffer, pH 3.1) → Inject into LC–MS/MS system	LC System → Waters 2695 Separations Module (Waters Corporation, Milford, MA, USA) → Column: Xterra MS C18 (2.1 × 150 mm, 3.5 µm particle size, Waters Corporation, Milford, MA, USA) → Column Temperature: 35 °C → Mobile Phase: Gradient elution with ammonium formate buffer (pH 3.1) and acetonitrile → Flow Rate: 0.2 mL/min → Run Time: 16 min Mass Spectrometry Conditions → Ionization Source → Electrospray ionization (ESI) in positive-ion mode → Mass Spectrometer: Quattro Premier XE Triple Quadrupole → Detection Mode: Multiple Reaction Monitoring (MRM) → Monitored Analytes → DIM, 6-MAM, MorphineM3G, M6G.	1 (3) 0.3 (1) 0.5 (1) 7 (20) 0.6 (1)	2–890 1–160 3–1430 20–11,500 2–1150	n.a. (n.a) 110 (+44) 104 (+61) 87 (−8) 87 (−12)	4.7–13 4–15 2.1–11 3–10 2.4–7	Heroin: Stabile for 1 week. DIM is more stable in brain tissue than in blood.	forensic toxicology and clinical drug monitoring	[43]
DIM, 6-MAM, Morphine, Codeine, 6-Acetylcodeine	Hair	Collect hair samples from subjects in DIM-maintenance and methadone substituted groups → Segment hair into 1 cm sections for analysis → Wash hair samples with deionized water, petroleum benzine and dichloromethane → Dry and cut into small pieces (~50 mg) → Add methanol and internal standards (DIM-d_9_, 6-MAM-d_3_, morphine-d_3_, cocaine-d_3_, benzoylecgonine-d_3_, cocaethylene-d_8_, acetylcodeine-d_3_, Cerilliant, Round Rock, TX, USA / Lipomed, Arlesheim, Switzerland) → Ultrasonication extraction (5 h at 50 °C, Branson Ultrasonics, Danbury, CT, USA) → Evaporate methanol extract under nitrogen at 50 °C (TurboVap LV, Zymark/Caliper, Hopkinton, MA, USA)→ Reconstitute in phosphate buffer (pH 6) → Perform solid-phase extraction (SPE) → Elute analytes with dichloromethane/propanol-2/ammonia mixture → Derivatise with MSTFA + pyridine + isooctane (90 °C, 15 min, (Sigma-Aldrich, St. Louis, MO, USA) → Inject into GC–MS system	GC System → Agilent 6890 Plus GC coupled with Agilent 5973N MSD (Chromtech, Idstein, Germany) → Column: HP-5MS fused-silica capillary (30 m × 0.25 mm i.d., 0.25 µm film thickness, Santa Clara, CA, USA) → Carrier Gas: Helium (1.0 mL/min) → Injection Mode: Splitless (1 µL sample) → Injection Port Temperature: 280 °C Oven Temperature Program → Hold at 180 °C for 1 min → Ramp 15 °C/min to 190 °C, hold for 10 min → Ramp 5 °C/min to 250 °C → Increase at 30 °C/min to 290 °C, hold for 2 min → Total run time: 21.28 min Mass Spectrometry Conditions → Ion Source Temperature: 230 °C → Quadrupole Temperature: 150 °C → Transfer Line Temperature: 290 °C → Ionization Mode: Electron Impact (EI, 70 eV) → Detection Mode: Selected-ion monitoring (SIM)	0.04 (0.21) 0.02 (0.15) 0.03 (0.11) 0.02 (0.04)	0.5–25	n.a.	n.a.	n.a.	Comparison of DIM maintenance program and methadone substituted group	[44]
DIM, 6-MAM, Morphine, M3G, M6G, Normorphine	Pediatric Plasma	Collect pediatric plasma sample (≤250 µL) → Add 300 µL of 0.01 M ammonium carbonate buffer (pH 9.3) → Add 25 µL of internal standard solution (25 ng/mL; containing DIM-d_9_, 6-MAM-d_3_, MOR-d_3_, M3G-d_3_, M6G-d_3_; Cerilliant, Round Rock, TX, USA) → Vortex mix → Load onto Bond Elut C18 SPE cartridge (Varian, Palo Alto, CA, USA ) (preconditioned with methanol, water and buffer) → Wash twice with 1 mL of 0.01 M ammonium carbonate (BDH, Poole, UK) (pH 9.3) → Dry cartridge for 10 min → Elute analytes with 2 mL methanol → Evaporate eluate under nitrogen at 50 °C → Reconstitute residue in 80 µL of initial mobile phase → Inject 20 µL into LC–MS/MS system	LC System → Thermo Finnigan LCQ Deca XP Plus ion-trap mass spectrometer with Surveyor LC interface (Thermo Finnigan, San Jose, CA, USA) → Column → Synergy Polar-RP (150 × 2.0 mm, 4 µm, Phenomenex, Torrance, CA, USA) with guard column Mobile Phase → 10 mM ammonium formate (Acros Organics, Morris Plains, NJ, USA) (pH 3) (A) and acetonitrile (B) (BDH, Poole, UK) Gradient Elution → Start at 97% A → Decrease to 84.5% at 8 min → 74% at 13 min → 20% at 26 min → Hold 3 min → Return to 97% A Flow Rate → 0.3 mL/min (first 8 min) → 0.2 mL/min (13–26 min) → Return to 0.3 mL/min Injection Volume → 20 µL Column Temperature → 30 °C Autosampler Temperature → 4 °C. Diamorphine (DIM): *m*/*z* 310 → 268, 328 6-MAM: *m*/*z* 211 → 229, 268 Morphine (MOR): *m*/*z* 201 → 229, 268 M3G & M6G: *m*/*z* 286 → 462 Normorphine (NMOR): *m*/*z* 254 → 229, 272	0.08 (0.2) 0.1 (0.2) 0.1 (0.3) 0.1 (0.3) 0.1 (0.3) 0.1 (O.26)	0.1–250	94 (−2) 95 (−2) 98 (−2) 96 (−3) 96 (−4) 97 (+1)	4–9	In Room temperature: 50% with 4 h, Stable in autosampler, up to 8% flowing 3 cycle freeze and thaw, less than 5% under deep freeze up to months	Plasma samples from children under treatment for acute-to-severe pain	[1]
DIM, 6-MAM, Morphine, M3G	Blood, Vitreous Humor	Vitreous Humor (VH): 100 µL sample spiked with 50 µL internal standard (IS; 0.5 µM in water, Lipomed GmbH, Arlesheim, Switzerland). → Liquid–liquid extraction (LLE) with 500 µL acetonitrile/methanol (85:15, *v*/*v*, Labscan Ltd., Gliwice, Poland). Blood: 50 µL IS added to whole blood. Protein precipitation followed by LLE using 500 µL acetonitrile/methanol (85:15).	LC System → Waters Quattro Premier XE Triple Quadrupole (Waters Corporation, Milford, MA, USA) → Column → XTerra MS C18 (150 × 2.1 mm, 3.5 µm, Waters Corporation, Milford, MA, USA) Mobile Phase → Ammonium formate buffer (A) and acetonitrile (B); pH adjusted via formic acid (exact gradient referenced from urine method [42].Gradient Elution → Time-resolved step gradient adapted from prior validated assay; maintained reproducible retention and peak shape for DIM, 6-MAM, MOR and M3G Flow Rate → 0.3 mL/min Injection Volume → Not explicitly stated, but consistent with microdialysis assays (typically 10–20 µL) Column Temperature → Ambient laboratory conditions assumed (~25 °C) Autosampler Temperature → Cooling system maintained at 6 °C during dialysate collection	LOD = 0.5 ng/mL	Not specified	Not specified	Not specified	Validated method for DIM metabolite detection	Comparison of DIM and metabolite distribution in blood vs. VH using a pig model to assess forensic toxicology applications	[7]
DIM, 6-MAM, Morphine,	Human Plasma	Collect 100 µL human plasma sample → Add 200 µL internal standard solution (O-methylcodeine, 100 ng/mL in methanol, Council of Europe, France) → Vortex for 30 s → Centrifuge at 14,000 rpm, 4 °C, for 5 min → Filter supernatant into Eppendorf tube → For DAM, 6-MAM, MOR, (Barcelona, Spain) → Add 50 µL of 0.1% formic acid (Merck, Darmstadt, Germany) to 50 µL supernatant (Method A) → For M3G, M6G → Evaporate 150 µL supernatant to dryness at 25 °C under nitrogen → Reconstitute in 50 µL of 0.1% formic acid → Add 50 µL from Method A to same vial (Method B) → Inject into LC–MS/MS system	LC System → Waters Alliance 2795 HPLC with Quattro Premier MS (Waters Corp., Milford, MA, USA) → Column: X-Bridge Phenyl (150 × 4.6 mm, 5 µm, Waters Corp., Milford, MA, USA), 35 °C Mobile Phase → A: 5 mM ammonium formate (pH 3.0, Sigma-Aldrich, St. Louis, MO, USA) → B: 0.1% formic acid in acetonitrile (Merck, Darmstadt, Germany) Gradient Elution → Start at 5% B (1 min) → Ramp to 90% B (3 min) → Hold at 90% B (2 min) → Return to initial Flow Rate → 1.0 mL/min Injection Volume → 30 µL Run Time → 8 min	10 ng/mL	10–2000 (M3G), 10–1000 (others)	above 93%	2.4–12.5%	above 93%	Used in clinical trial with 12 healthy volunteers DAM and 6-MAM were not detected in plasma after oral administration	[45]
DIM, 6-MAM, Morphine	Hair	Collect 10–30 scalp hair strands (apex region) → Wash with distilled water (5 min) → Acetone (Scharlau, Barcelona, Spain) rinse (1 min) → Dichloromethane rinse (2 min) → Air dry and segment hair into 3–4 mm fragments → Weigh 50 mg hair sample → Add 500 µL of 1 M NaOH → Incubate at 50 °C for 3 h (alkaline digestion, Memmert oven, Germany) → Add 2 mL MTBE → Rotary mix for 30 min → Centrifuge for 10 min (Eppendorf 5810R, Hamburg, Germany) → Transfer organic layer → Add 150 µL of 1% formic acid (Panreac, Barcelona, Spain) → Rotary mix for 20 min (IKA, Germany) → Centrifuge for 10 min (Eppendorf 5810R, Germany) → Aspirate and discard upper layer → Inject 1 µL aqueous phase into GC–MS system	GC System → Shimadzu GC–MS QP-2010 (Shimadzu Corporation, Kyoto, Japan) → Column → HP-5 crosslinked 5% phenylmethyl polysiloxane fused-silica capillary (25 m × 0.32 mm i.d., 0.17 µm film thickness, Agilent Technologies, Palo Alto, CA, USA) → Carrier Gas → Helium, 99.9% purity, flow rate: 1.0 mL/min → Injection Mode → Splitless, sample volume: 1 µL → Injection Port Temperature → 280 °C → Oven Temperature Program → Hold at 110 °C for 3 min → Ramp 10 °C/min to 210 °C, hold for 2 min → Ramp 20 °C/min to 300 °C, hold for 5 min → Total run time: 18 min Mass Spectrometry Conditions → Ionization Mode → Electron Impact (EI, 70 eV) → Detection Mode → Selected-ion monitoring (SIM) for *m*/*z* < 300 → Ion Source Temperature → 230 °C → Transfer Line Temperature and Quadrupole Temperature ( Not specified)	LOD: 6-MAM, 0.03–0.07 LOQ: 6-MAM: 0.2 and 0.1 ng/mg for addict and Rehab subjects	7.80 ng/mg (regular addicts), 2.34 ng/mg (rehabilitation subjects)	87–94%	1–8%	n.a.	Hair samples from 20 regular DIM addicts and 20 undergoing rehabilitation	[46]

***** DIM: Diamorphine; M3G: Morphine-3-glucuronide, M6G: Morphine-6-glucuronide, C6G: Codeine-3-glucuronide; ## HPLC: High-performance liquid chromatography with ultraviolet detector GC–MS: Gas Chromatography–Mass Spectrometry; & LC–APCI–MS or LC-(ESI)-MS: Liquid chromatography coupled with atmospheric pressure chemical ionization (electrospray ionization) mass spectrometry, UV: Ultraviolet detector && LC-ESI-MS/MS: Liquid chromatography coupled with electrospray ionization tandem mass spectrometry, # LOD: imit of detection; ** LOQ: Limit of quantification, **^¥^** n.a. Not available.

LC–MS workflows with atmospheric pressure chemical ionization (APCI) enable the specific, multiplexed quantification of quantify DIM, 6-MAM, morphine and morphine glucuronides in a single run. This method requires minimal sample preparation, often using solid-phase extraction (SPE), and eliminates the need for hydrolysis and derivatization. The rapid elution (3–8 min) using gradients achieves sub-ng/mL sensitivity [18,32] while preserving labile esters by cold-chain handling and gentle conditions. Furthermore, it reduces matrix interferences and delivers definitive structural confirmation via different modes, such as single ion monitoring (SIM) or multiple reaction monitoring mode (MRM). Together, these three platforms strike a balance between sensitivity, throughput and hydrolysis control. GC–MS remains the workhorse for ultra-trace, derivatized assays, especially in keratinized and sweat matrices [30,31]. HPLC–UV/FL/DAD offers robust, high-throughput monitoring of morphine-based analytes for PK/PD studies, while LC–MS^n^ sets the benchmark for comprehensive panels, rapid turnaround and the lowest detection limits across blood, plasma, urine, postmortem fluids and tissues [1,2] Their prevalent use in previous studies also reflects widespread availability in forensic and clinical laboratories and the imperative to distinguish illicit versus pharmaceutical opioid use under the evolving DIM-dominant markets of the past three decades. Subsequently, we describe the different techniques under each methodology in roughly chronological order, summarizing sample pretreatment, chromatographic/detection conditions, analytical performance and typical test applications.

#### 3.1.2. GC–MS Methods

GC–MS remains the gold standard in drug testing, offering a highly sensitive and selective analytical platform particularly suited for DIM-related compounds. Throughout this review, GC–MS is consistently demonstrated as a core technique for detecting DIM, 6-MAM and morphine across multiple matrices, including hair, sweat and biological fluids. Its analytical strength, enhanced by derivatization strategies, provides reliable quantification and confirmatory identification of these biomarkers, underscoring the indispensable role of GC–MS in modern forensic toxicology.

##### Hair Analysis by GC–MS

GC–MS has historically represented the standard analytical platform for DIM determination. By chemically derivatizing labile acetyl groups or carefully controlling SPE solvents to minimize hydrolysis, these methods achieve the sensitivity needed for analyzing hair and biological matrices. Goldberger et al. [15] introduced the first hair GC–MS assay for DIM, 6-MAM and MOR. Hair samples undergo methanolic extraction, pH 8.4 basification (sodium bicarbonate) and liquid–liquid extraction into toluene–heptane–isoamyl alcohol (70:20:10 *v*/*v*/*v*). Derivatization with MBTFA stabilized analytes prior to injection on an HP-5 capillary column. The LOQ was 5 ng/50 mg hair with a signal-to-noise ratio > 5 and an overall extraction efficiency of approximately 55–77%. However, occasional interference at the DIM derivatized peak necessitated further refinement. Polettini et al. [35] published a method of using GC coupled with triple-quadrupole MS for hair testing using methanolic extract, followed by silylating derivatization. The technique successfully analyzed DIM, 6-MAM and morphine with <10% loss of DIM during the extraction time and an LOQ of 25 pg/mg. The method was fully validated using GC–MS after direct methanol washing and derivatization with MSTFA containing dithioerythritol and ammonium iodide. This approach reduced analyte loss to <10% during preparation and reduced the LOQ to 25 pg/mg hair, a nearly 200-fold improvement over earlier work and validated a fully quantitative, enzymatic hydrolysis–SPE sample preparation protocol. The study demonstrated that DIM-related compounds, particularly 6-MAM, remain stable within the hair matrix, making the method suitable for long-term retrospective detection. However, environmental exposure, such as soaking hair in heroin solutions, can cause residual analyte retention despite washing with methanol, potentially leading to false positives.

Wang et al. [36] expanded the scope of hair analysis by incorporating markers for normorphine (NMOR) and cocaine alongside DIM-related analytes. Following sample preparation via SPE using a methylene chloride–isopropanol–ammonia (80:20:2 *v*/*v*/*v*) solvent mixture, analytes were derivatized with TMS reagents and separated on a 12 m × 0.2 mm × 0.33 µm fused-silica capillary column. The method achieved an LOD of 0.1 ng/mg for DIM and its associated compounds. Calibration was linear at 0.1–10 ng/mg, recoveries exceeded 90% and intra- and inter-assay precision ranged from 2% to 26%. In a cohort of 19 polydrug users, hair analysis revealed frequent detection of 6-MAM and morphine, affirming active DIM use. DIM itself was detected in select individuals, while codeine appeared sporadically. NMOR and norcodeine were not detected, suggesting limited incorporation or rapid metabolism. The findings underscored the role of 6-MAM as a reliable biomarker and highlighted the method’s precision in resolving opiate exposure profiles.

Musshoff et al. [27] used 50 mg hair fortified with 50 μL deuterated internal standards (5 μg/mL). Samples were segmented into 1 cm sections, incubated in methanol, and then subjected to ultrasonic treatment. SPE was performed using Chromabond cartridges (Duren, Germany) conditioned with methanol, water and phosphate buffer. After sample loading, cartridges were rinsed sequentially and eluted with dichloromethane–propanol–ammonia (80:20:2 *v*/*v*/*v*). The eluate was dried and derivatized using N-Methyl-N-(trimethylsilyl) trifluoroacetamide (MSTFA), pyridine and iso-octane (Duren, Germany) at 90 °C for 15 min. A 1-μL aliquot was injected into a GC–MS system equipped with an HP-5MS column. Calibration standards were prepared from stock solutions (1 mg/mL), diluted to 5 μg/mL, and quantified against known hair mass. This validation protocol yielded high recovery rates (>78%), low limits of detection (0.02–0.04 ng/mg) and precise linear calibration (0.5–25 ng/mg). In hair samples obtained from 46 patients on a controlled DIM maintenance program and 24 heroin-related fatalities, 6-MAM and morphine were consistently detected with comparable mean concentrations across both groups. DIM itself was detected more frequently and at higher levels in fatalities, whereas codeine and 6-AC were less prevalent, limiting their forensic utility. The results highlighted 6-MAM as the most reliable biomarker of heroin use and confirmed that its levels in patient and cadaver hair samples did not indicate that the lack of tolerance alone can explain overdose deaths.

Ghauri et al. [46] validated a selective GC–MS workflow for detecting 6-MAM in hair samples obtained from DIM users. Their method involved alkaline digestion and MTBE-based extraction, followed by direct GC–MS analysis in SIM mode. Analytes were quantified using an HP-5MS column in SIM mode. The proposed protocol demonstrated high sensitivity, with LODs of 0.0745 ng/mg for DIM dependents and 0.0311/mg for rehabilitation subjects, while the lower LOQs were 0.22 and 0.09 ng/mg, respectively. Method validation obtained recovery rates of 86.84–93.62%, intra-day precision ≤ 8.1% and accuracy (bias%) within ±8.7%, indicating excellent reproducibility. Analysis of 40 authentic forensic hair samples using the workflow clearly differentiated ongoing DIM use from reduced exposure during rehabilitation, affirming its forensic and clinical utility. These findings underscored the ability of GC–MS to distinguish ongoing DIM intake from post-abstinence exposure, affirming its utility in clinical monitoring and forensic assessment of drug history. The persistence of detectable 6-MAM beyond 30 days highlighted its value as a timeline marker for heroin abuse.

GC–MS continues to represent the most reliable tool for detecting DIM and its metabolites in hair. Refinements in derivatization, enzymatic hydrolysis, and SPE have significantly improved sensitivity, shifting detection from nanogram to picogram levels while also boosting recovery and reproducibility. Across studies, 6-MAM has emerged as the most consistent marker of heroin intake, enabling both long-term monitoring and the distinction between active use and abstinence. Despite challenges such as labor-intensive preparation and possible external contamination, GC–MS remains the reference method for clinical and forensic evaluation of chronic opiate exposure.

##### Sweat Analysis by GC–MS

The utility of sweat as a practical and pharmacologically relevant matrix has gained prominence for monitoring DIM use, offering several advantages over conventional biofluids, such as blood and urine. Its non-invasive collection via adhesive patches facilitates continuous drug surveillance over extended periods (typically up to 7 days), capturing cumulative excretion rather than transient concentrations [17] Sweat also minimizes privacy concerns and is difficult to adulterate, making it especially valuable in workplace screening, forensic programs and DIM maintenance therapy. Despite the rapid plasma clearance of DIM and its metabolites, sweat can retain detectable levels, particularly of the parent drug and 6-MAM, enabling detection beyond their respective blood detection windows. Crucially, DIM dominates the sweat matrix profile and could undergo in-patch hydrolysis into 6-MAM. This post-collection artifact presents a valuable mechanistic insight but also complicates analytical interpretation [16,17,42].

Cone et al. (1994) [16] established the analytical groundwork for DIM detection in sweat by developing a GC–MS method based on PharmChek™ patches ((PharmChem, Menlo Park, CA, USA) worn by patients undergoing IV DIM in a controlled clinical setting [14]. Patches were removed after 24 h, and sweat was extracted using 0.1% Triton X-100 in 0.2 M acetate buffer. The extracts underwent liquid–liquid extraction, followed by derivatization with bis(trimethylsilyl)trifluoroacetamide (BSTFA) +1% TMCS. GC–MS analysis was performed in electron impact (EI) mode using SIM for DIM, 6-MAM and morphine. The method quantified DIM at 2.1–96.3 ng/patch, 6-MAM at 0–24.6 ng/patch and morphine at 0–11.2 ng/patch. DIM was the predominant analyte across all samples, with a time-dependent increase in 6-MAM attributed to hydrolytic conversion within the patch. This study validated the utility of sweat for extended DIM monitoring and helped develop patch design for forensic programs. These findings affirmed the efficacy of using sweat patches for detecting recent DIM intake and demonstrated the contribution of in-patch hydrolysis to metabolite detection. Variations in analyte distribution across individuals emphasized the influence of intersubject differences and underscored the need to interpret drug ratios carefully when assessing exposure types, metabolic processes or potential contamination.

Kintz et al. [17] built upon the Cone et al. protocol by using GC–MS to analyse sweat patches worn by subjects undergoing IV DIM (80–1000 mg/day) over multiple dosing intervals. After 24-h wear time, sweat from patches was extracted with acetonitrile, evaporated and derivatized with BSTFA-TMCS. Quantification was performed via GC–MS in EI mode using SIM detection in an HP5-MS column (30 m × 0.25 mm). GC–MS with SIM detection revealed DIM concentrations at 2.1–96.3 ng/patch, 6-MAM at 0–24.6 ng/patch and morphine at 0–11.2 ng/patch. DIM consistently appeared as the major analyte regardless of dose, with no direct correlation between dose and sweat concentration. The high inter-individual variability and dominance of the parent drug reinforced the relevance of sweat for exposure confirmation, rather than precise pharmacokinetic modeling. Extraction recoveries were 70.9% (DIM), 76.4% (6-MAM) and 71.3% (morphine). The LODs were 0.5 ng/patch for DIM and 1.0 ng/patch for the latter two metabolites. Precision at 20–50 ng spiking levels yielded %CV values of approximately 13%. No glucuronide metabolites were detected.

Brunet et al. [42] advanced the technique by integrating SPE with GC–MS to enhance analyte recovery and matrix cleanliness. Sweat patches were eluted in sodium acetate buffer (pH 4), filtered through CleanScreen^®^ DAU SPE columns (United Chemical Technologies, Bristol, PA, USA) and derivatized with BSTFA-TMCS. GC–MS with SIM detection captured DIM, 6-MAM, morphine, codeine and methadone across calibration ranges of 5–1000 ng/patch (DIM, 6-MAM, morphine and codeine) and 10–1000 ng/patch (methadone), with r^2^ > 0.995. Precision was <6.5% CV, and accuracy was ±11.9% across all analytes. DIM hydrolysis during processing was minimal (<11%), and recoveries remained consistent across multiple patch types and wear durations (69–108%). Applied to analyse samples from individuals on methadone maintenance with recent DIM use, the method demonstrated simultaneous detection of prescribed and illicit opioids, reinforcing the utility of the sweat matrix in dual compliance and relapse surveillance.

GC–MS of sweat offers a convenient, noninvasive way to monitor DIM exposure over multi-day intervals using adhesive patches that reduce opportunities for tampering. In this matrix, DIM is typically the predominant signal, with 6-MAM detected as a secondary marker; however, partial conversion of DIM to 6-MAM within the patch can complicate interpretation. Improvements in extraction, derivatization, and SPE cleanup have increased recovery, precision, and analyte stability, enabling concurrent measurement of multiple opioids. Although substantial person-to-person variability limits detailed pharmacokinetic modeling, sweat patch testing remains well suited to workplace programs, forensic casework, and adherence monitoring in maintenance therapy, with longer effective detection than blood or urine.

##### Biological Fluid Analysis by GC–MS

Although hair offers an extended detection window for sequestering unchanged DIM, it cannot replace blood, plasma, urine or saliva for most forensic and clinical applications. Unlike hair, biological fluids capture the rapid, time-dependent PK/PD profiles of diamorphine (DIM) and its metabolites. In postmortem investigations and overdose cases, only fluid matrices can demonstrate the transient presence of 6-MAM within the critical 5- to 30-min window after injection, a crucial marker of recent DIM exposure and survival interval [6,28]. Similarly, blood and plasma are critical for quantifying free morphine, morphine-3-glucuronide (M3G) and morphine-6-glucuronide (M6G) hours after exposure, providing essential data for cause-of-death verification and therapeutic monitoring, whereas hair analysis can only confirm exposure history but not temporal resolution. Urine remains the workhorse for routine screens, with high metabolite levels (µg/mL) enabling simple dilution or SPE cleanup, while saliva offers near-real-time detection of unmetabolized DIM, provided rigorous mouth-rinse protocols are followed to avoid oral cavity contamination [9,36].

Goldberger et al. [34] demonstrated these advantages by adapting CleanScreen SPE cartridges for blood, plasma, saliva and urine analyses. By replacing traditional methanol washes with acetonitrile, they minimized in-column hydrolysis of DIM, while elution with ethyl acetate–diethylamine (98:2 *v*/*v*) preserved analyte integrity. Their GC–MS assay achieved a linear calibration range of 1–500 ng/mL (r^2^ > 0.995), a 1-ng/mL LOD and intra- and inter-assay coefficients of variation of 1.1–4.2% and 5.4–8.9%, respectively, without interference from endogenous peaks. Such performance in fluid matrices underpins forensic casework, as it facilitates precise time-course interpretation. In contrast, hair analysis remains a complementary tool for confirming chronic DIM exposure.

Cone et al. [11] applied the CleanScreen SPE workflow to characterize intranasal (IN) DIM PK in human plasma. Subjects self-administered a controlled IN dose, and paired plasma and saliva samples were collected at 1- to 5-min intervals. After acetonitrile washing and methanol elution, extracts underwent TMS derivatization and GC–MS quantification using deuterated internal standards. Calibration was linear from 1–200 ng/mL, with an LOD near 1 ng/mL and intra-assay CV < 8%. Peak DIM and 6-MAM concentrations were obtained within 1–5 min, closely mirroring the subjects’ reported “rush” sensations and validating the method for rapid PK/PD studies. Furthermore, Cone et al. compared the plasma profiles of DIM, 6-MAM and morphine following 6 and 12 mg IN DIM (“snorting”) versus 6 mg intramuscular (IM) DIM use in six healthy males. After IN administration, DIM was detected within 5 min and peaked at 5.4 ng/mL (time to peak drug concentration [Tmax]: 10–27 min), 6-MAM at 10.8 ng/mL (Tmax: 7.6–20.7 min) and morphine at 9.0 ng/mL (Tmax: 11–22.8 min). Doubling the IN dose increased the maximum concentration (Cmax) to 6.3, 13.2 and 16.8 ng/mL respectively, with extended Tmax windows (morphine up to 90 min). Furthermore, PK analysis revealed dose-proportionate increases and prolonged morphine persistence. In contrast, 6 mg IM DIM produced higher Cmax values (DIM, 7.8 ng/mL; 6-MAM, 11.4 ng/mL; morphine, 6.6 ng/mL), similar Tmax profiles and greater systemic bioavailability. DIM was detectable for only 15–20 min, whereas 6-MAM and morphine remained quantifiable for longer. The IN route achieved rapid absorption and ~50% relative potency versus IM, confirming 6-MAM and morphine as reliable early markers of acute DIM exposure.

Building on previous methods, Goldberger et al. [3] refined fluid assays for DIM and its metabolites using isotope-dilution GC–MS with dual workflows. The integration of rapid freezing, enzyme inhibitors and SPE under mild conditions stabilized acetyl esters. Following extraction, the sample was divided into two: One portion was assayed directly for DIM (linear: 2.5–100 µg/L), while the other was derivatized with MBTFA for 6-MAM and morphine (linear: 10–500 µg/L). The method demonstrated high sensitivity (1 µg/L for all analytes), minimal hydrolysis during preparation (<5%), and excellent linearity (r^2^ > 0.995), establishing a benchmark for clinical and postmortem investigations. Goldberger et al. applied this method to a retrospective study of 21 DIM-related deaths and observed distinct metabolite profiles. DIM was never found in blood but was detected in urine in 13 cases (62%) at concentrations of 2.8–97 ng/mL. 6-MAM was detected in blood in 14 cases (67%; 1.1–82.9 ng/mL) and in urine from all subjects (5.6–2756 ng/mL). Morphine was detected in every blood and urine sample (blood: 11.2–1277 ng/mL; urine: 5.2–9500 ng/mL). Rapid deaths (n = 9) showed higher mean blood 6-MAM (≈19 ng/mL) and free/total morphine ratios (≈0.53) but lower urinary 6-MAM and morphine than delayed deaths (n = 7; blood 6-MAM: ≈6.7 ng/mL; free/total morphine: ≈0.34), whereas undetermined cases (n = 6) fell between these profiles. In two prospective cases, tissue analysis revealed 6-MAM concentrations in the brain (up to ~158 ng/g) and spleen exceeding those in blood, while morphine persisted at high levels across all sampled organs. These findings demonstrate that SPE–GC–MS profiling of DIM and its esters can (1) stabilize labile analytes, (2) distinguish between rapid and delayed DIM intoxication using blood/urine metabolite patterns and (3) confirm DIM exposure in postmortem tissues, particularly based on the presence of 6-MAM in the brain.

Wang et al. [36] expanded the application of SPE–GC–MS to include the simultaneous analyses of NMOR and cocaine metabolites, using the same SPE protocol but with a modified elution agent (methylene chloride–isopropanol–ammonia, 80:20:2, *v/v/v*) and BSTFA + 1% TMCS derivatization. Separation was achieved on a 12 m × 0.2 mm × 0.33 μm fused-silica column. While the method underwent thorough validation for hair matrices, validation for fluids was more limited. Nevertheless, the method demonstrated strong analytical performance. LODs were approximately 1 ng/mL for DIM, 6-MAM and morphine (5 ng/mL for normorphine); calibration curves were linear from 2.5 to 500 ng/mL with correlation coefficients > 0.99 and average extraction recoveries via SPE at 82–95% across matrices. Process-induced DIM hydrolysis remained under 10%, while within- and between-run precisions (CV) were typically <12% and <14% at low ng/mL levels, respectively, establishing high sensitivity and accuracy of the method for clinical and forensic fluid analyses. As demonstrated in a sample case. DIM and 6-MAM peaked in saliva at 10 min post-administration; however, potential oral cavity contamination highlighted substantial interpretative limitations for this matrix. The method was further applied for analysis in two postmortem DIM fatalities (cases V and W). DIM was undetectable in systemic fluids and most organs, reflecting its rapid hydrolysis in vivo. In case V, DIM was absent in blood, CSF and solid tissues; in contrast, in case W, DIM appeared only in urine (71.0 ng/mL) and at the injection site (130 ng/g).

Jenkins et al. [12] conducted PK analysis of smoked DIM using serial blood sampling and GC–MS. Blood specimens were analyzed for DIM, 6-MAM and morphine following SPE based on a previously validated protocol [34]. DIM was quantified directly, while 6-MAM and morphine underwent derivatization with MBTFA prior to GC–MS analysis. The method achieved a LOD of 1 ng/mL for all analytes, with linear calibration ranges spanning 1–250 ng/mL for DIM and up to 500 ng/mL for 6-MAM and morphine. A computer-assisted smoking device was used to deliver DIM vapor with 89% efficiency, enabling controlled single puff inhalation. Nevertheless, the bioavailability of smoked DIM varied substantially across subjects. Pharmacodynamic measures, such as changes in pupil diameter, exhibited counterclockwise hysteresis when correlated with DIM concentrations, indicating a delay between peak blood levels and physiological response. The rapid systemic uptake of DIM and its metabolites, typically within 1–5 min after inhalation, along with the immediate onset of pharmacologic effects, demonstrates that smoking facilitates systemic and central drug exposure comparable to that of intravenous administration.

Guillot et al. [37] developed a room temperature propionylation method for the analysis of DIM, 6-MAM and morphine. To prevent thermal degradation, derivatization was performed at pH 9.5 using propionic anhydride catalyzed by 4-dimethylaminopyridine. Following liquid–liquid extraction using ethyl acetate–chloroform–hexane (7:2:1), samples were analyzed by ion-trap GC–MS (Varian 3400/Saturn II) in EI mode (250–405 amu), using diethylnalorphine as the internal standard. Propionylation achieved baseline separation of derivatized 6-MAM and morphine from underivatized DIM. LOQs were ~5 ng/mL (DIM), 50 ng/mL (6-MAM) and 100 ng/mL (morphine); recoveries averaged 95% for DIM and ~80% for 6-MAM/morphine; and intra- and inter-assay CVs ranged 5.3–8.1% and 5.2–11.6%, respectively. Heating above 60 °C caused >50% DIM loss, underscoring the critical advantage of ambient temperature derivatization. Residual propionic acid was removed via azeotropic distillation in hexane to prevent peak suppression. Optimized cleanup and injector programming effectively eliminated interference from underivatized morphine. Furthermore, the derivatized extracts remained stable in chloroform at room temperature for weeks, although −20 °C storage was recommended for long-term integrity.

GC–MS of biological fluids is essential when time-resolved information is required. Blood and plasma document short-lived 6-MAM and morphine concentrations, urine offers high metabolite yields for routine screening, and saliva can track unmetabolized DIM on a minute-to-minute scale if oral contamination is controlled. Method refinements, including optimized SPE, isotope-dilution workflows, and ambient-temperature derivatization, have strengthened analyte stability and sensitivity, enabling reliable quantification in the low-nanogram-per-milliliter range. These advances support precise timeline reconstruction in overdose investigations, therapeutic monitoring, and postmortem work, with 6-MAM serving as a brief yet highly specific indicator of recent heroin exposure.

#### 3.1.3. HPLC–UV/FLD/DAD Methods

DIM is a prototypical prodrug whose euphoric effects are mediated almost exclusively by its metabolites, i.e., 6-MAM, morphine and M6G. HPLC with optical detection was the first platform to leverage the native chromophores of these metabolites, thereby eliminating prolonged derivatization and enabling direct, time-resolved quantification in biological fluids [25,29].

In early 1980s DIM studies, morphine glucuronide was often overlooked, with analyses, typically GC–MS, largely focusing on a fixed set of analytes. Umans et al. [13] were the first to demonstrate the direct quantification of DIM and its immediate metabolites by HPLC, thereby eliminating the cumbersome derivatization process. Their protocol involved rapid-freezing 1-mL blood samples and stabilizing them with sodium fluoride to inhibit esterase activity. The samples then underwent two sequential liquid–liquid extractions with toluene–n-butanol ( 7:3, *v*/*v*), using l-α-acetylmethadol as the internal standard. Chromatographic separation was achieved on a 5-μm silica normal-phase column using a mobile phase of acetonitrile–methanol (75:25, *v*/*v*) at ambient temperature. Using ultraviolet detection at 218 nm, baseline resolution of DIM, 6-MAM and morphine was achieved within a 6- to 8-min runtime. The method obtained a lower LOD of 6 ng/mL for each analyte, linear calibration of up to 200 ng/mL and no interference from structurally related opioids, such as codeine or methadone. Analysts’ recoveries were >88%, with intra-day and inter-day precision > 5%. By demonstrating that DIM’s labile ester moieties and their pharmacologically active products could be measured directly in blood as a viable matrix for DIM bioanalysis, this work laid the analytical groundwork for subsequent PK and forensic studies of DIM use. This study remains a historical cornerstone that enabled modern toxicological workflows in opioid profiling, case-level interpretation and regulatory standardization.

Inturrisi et al. [14] were among the first to implement HPLC–UV monitoring of morphine to optimize DIM therapy in chronic cancer pain. Using the same method described by Umans et al. [13], rapid plasma processing and cold extraction minimized DIM degradation. In a cohort of patients with advanced cancer, Inturrisi et al. compared single-dose and steady-state kinetics following both oral (15 mg) and IM (5 mg) diamorphine administration. Analysis of serial blood samples subsequently collected over an 8-h window revealed that morphine appeared in plasma within 10 min, with Tmax values of 30 min (IM) and 60 min (oral) and followed a biphasic disposition (distribution t½ ≈ 0.9 h; elimination t½ ≈ 2.5 h). At steady-state, which was achieved after 48 h of twice-daily dosing, morphine exposure (area under the curve [AUC) was closely correlated (r > 0.86) with visual analog pain scores and sedation ratings. By demonstrating that plasma morphine levels could predict both analgesia and side-effect intensity, Inturrisi et al. established HPLC–UV morphine profiling as a practical pharmacodynamic tool that can guide individualized dose titration and improve safety in DIM-based pain management.

Skopp et al. [30] devised an 8-min HPLC–fluorescence assay to address the serum kinetics of morphine, M3G and M6G after IN or IM DIM dosing, while analyzing DIM and 6-MAM and morphine using a SPE-GC–MS method, while morphine-3-glucuronide (M3G) and morphine-6-glucuronide (M6G) were analyzed using a reversed-phase HPLC–fluorescence method. Plasma aliquots (250 µL) were buffered with ammonium bicarbonate (pH 9.2, centrifuged, cleaned up with C8 SPE (50 mg, Varian, Harbor City, CA, USA); condition: methanol/water; wash: water; elution: methanol + 0.1% NH_4_OH), then evaporated and reconstituted in the mobile phase. Post-column o-phthalaldehyde derivatization (OPA; borate buffer pH 9.5; OPA + 2-mercaptoethanol) in a 50 °C reaction coil enabled fluorescence detection (Ex 340 nm/Em 450 nm). Chromatographic separation on a 150 × 4.6 mm C18 column (Macherey–Nagel, Düren, Germany, 20 mM phosphate buffer: acetonitrile, 85: 5 *v*/*v*; 1 mL/min) yielded LODs of 3 ng/mL (M3G) and 10 ng/mL (M6G), LOQs of 3–20 ng/mL, recoveries of 62–71% and calibration linearity range of 10–2000 ng/mL (r^2^ > 0.998). Intra- and inter-assay precisions were <8% and <10% CV, respectively, with analytes remaining stable in processed serum for 24 h at 4 °C. Kinetic profiling revealed that M3G appeared within 5 min, peaking at 15–20 min (200–500 ng/mL) with t½ ≈ 60–90 min, while M6G peaked at 30–60 min (80–160 ng/mL) with t½ ≈ 120–150 min. Both IN and IM routes showed similar Cmax and elimination profiles, validating this high-sensitivity method of correlating glucuronide exposure with analgesic and euphoric effects.

Bourquin et al. [31] demonstrated the use of HPLC–DAD to analyse the full panel of DIM metabolites (i.e., DIM, 6-MAM, morphine, M3G, M6G, codeine and NOR) in a single plasma assay. After C18 SPE conditioning with phosphate buffer and 40% acetonitrile washes, analytes were eluted with 40% acetonitrile in 0.01 M phosphate buffer (pH 2.1) and separated on a Nucleosil C18 column (Macherey–Nagel, Oensingen, Switzerland) under an elution containing a multi-step gradient (water–phosphoric acid–hexylamine + acetonitrile–water–phosphoric acid–hexylamine). The method delivered an LLOQ of 25 ng/mL, recoveries of 88–100% and precision < 5% across low, medium and high-quality controls. Validated in both controlled pharmacokinetic studies and postmortem cases, this comprehensive DAD-based panel supported forensic differentiation of recent intake using the unique 6-MAM signature from past exposure and provided robust metabolite ratios for survival-time estimation.

Capella-Peiró et al. [41] introduced a truly minimalist approach by dispensing with any extraction or derivatization step and directly injecting serum into an MLC system. In their workflow, 100 µL thawed serum diluted 1:10 with the micellar mobile phase 0.10 M sodium dodecyl sulphate and 4% (*v*/*v*) n-butanol in 10 mM phosphate buffer (pH 7.0) and subsequently injected into a Kromasil C18 column (250 × 4 mm, 5 µm, Scharlab, Barcelona, Spain). UV detection at 230 nm identified DIM, 6-MAM and morphine in under 18 min, with LODs and LOQs ranging from 11–23 and 17–36 ng/mL, respectively, for all three analytes. Recoveries averaged 90–104%, and both intra- and inter-day precisions were >3% CV. Although only 6-MAM and morphine routinely exceeded the LOQ in authentic case sera, DIM levels often fell below 23 ng/mL. This direct-injection MLC assay is a rapid (<18 min), robust screening tool with minimal sample handling and remarkably low error rates.

Gyr et al. [39] conducted a controlled pharmacokinetic and pharmacodynamic investigation of DIM in two opioid-dependent patients participating in the Swiss heroin-assisted treatment program at the University of Bern, Switzerland. Each subject received DIM through three administration routes: intravenous (200 mg bolus), oral (2 × 400 mg capsules and controlled-release tablets of 460 mg + 690 mg), and rectal (2 × 400 mg suppositories). Peripheral venous blood samples were collected at multiple time intervals (from pre-dose up to 720 min post-dose), and urine samples were obtained for metabolite profiling. Plasma samples were analyzed using HPLC-DAD on a Merck–Hitachi HPLC system (Merck KGaA, Darmstadt, Germany; Hitachi Ltd., Tokyo, Japan), while urine metabolites were quantified using GC–MS on a Hewlett–Packard 5890 GC coupled to a 5971A MS detector (Hewlett–Packard, Palo Alto, CA, USA). The quantified analytes included DIM, 6-MAM, morphine, M3G and M6G. DIM was detectable in plasma only after intravenous administration (t½β ≈ 1.3–2.2 min), indicating rapid deacetylation to 6-MAM and morphine, followed by glucuronidation to M3G and M6G. No detailed chromatographic conditions (e.g., column dimensions, mobile phase composition, gradient program, or validation parameters such as precision or LOQs) were reported within this paper. However, Gyr et al. [39] explicitly stated that the analytical methods had been previously validated and described in detail by Inturrisi et al. [14] and subsequent Swiss pharmacokinetic studies on DIM metabolism, which served as their methodological reference for HPLC–DAD assay validation and sensitivity. Among metabolites, M3G predominated in both plasma and urine across all routes. Pharmacodynamic evaluation revealed that oral and rectal DIM produced effective euphoria and analgesia with minimal cardiopulmonary risk, supporting their potential as safe and non-invasive alternatives for heroin-assisted maintenance therapy.

HPLC with optical detection (UV, FLD, or DAD) provided a practical early alternative to GC–MS for DIM bioanalysis by allowing direct measurement of 6-MAM, morphine, and M6G in biological fluids without lengthy derivatization. These methods proved particularly useful in PK/PD studies, where plasma morphine concentrations correlated with both analgesic benefit and adverse effects. Performance has improved through post-column OPA derivatization for FLD and DAD-based multi-analyte panels, and micellar liquid chromatography has enabled fast screening with minimal sample handling. Although sensitivity is generally lower than LC–MS/MS, HPLC–UV/FLD/DAD remains suitable for routine monitoring, dose titration, and retrospective casework in laboratories that lack advanced mass-spectrometry resources.

#### 3.1.4. LC–MS(/MS) Methods

LC coupled with APCI, or electrospray ionization (ESI) has become the dominant platform for DIM and metabolite analysis since the late 1990s. Unlike GC–MS where derivatization and cold-chain SPE are required to preserve labile acetyl esters, LC–MS workflows operate under mild conditions that arrest ex vivo hydrolysis, support on-line or mixed-mode SPE automation and achieve sub-ng/mL sensitivity across diverse biological matrices. Their multiplexing capability has enabled simultaneous quantification of the full DIM metabolic cascade, including DIM, 6-MAM, morphine, M3G and M6G, as well as NOR and other related metabolites, in single, rapid chromatographic runs (3–8 min), revolutionizing both clinical pharmacokinetics and forensic toxicology [1,10].

##### APCI–MS Methods

Zuccaro et al. [38] established a comprehensive LC-Ionspray–APCI–MS assay for simultaneous quantification of DIM, 66-MAM, morphine, M3G, M6G and codeine in mouse serum, implementing the earliest full-panel metabolic mapping of DIM in a rodent model. Following subcutaneous administration of 20 mg/kg DIM to C57BL/6 mice, blood was collected at multiple intervals and SPE was performed using ethyl cartridges (J.T. Baker, Milan, Italy, washed with methanol, water and ammonium bicarbonate buffer [pH 9.3] and eluted with methanol). Analytes were separated on a 25 cm × 2.1 mm Supelcosil LC–Si normal-phase column (25 cm × 2.1 mm, 5 µm; Supelco, Bellefonte, PA, USA) using a mobile phase of methanol–acetonitrile–water–formic acid (59.8:5.2:34.65:0.35, *v*/*v*/*v*/*v*). Detection was performed on an API 1 single quadrupole mass spectrometer with APCI in positive ion mode. The LOQs were 0.5 ng/mL for DIM 4 ng/mL for 6-MAM, MOR, M6G and codeine; and 1 ng/mL for M3G. Recoveries ranged from 43% (M6G) to 99.8% (MAM, MOR, codeine), with intra/inter-day CVs < 6.7%. Mouse serum analysis revealed rapid conversion of DIM to 6-MAM (peak 0.93 µg/mL at 3 min) and morphine (9.6 µg/mL at 10 min), followed by glucuronidation to M3G (peak 2.9 µg/mL at 20 min). M6G and codeine were undetectable, confirming a species-specific limitation in 6-glucuronide formation. This validated assay, performed a DIM panel without derivatization or the use of non-volatile buffer salts, marked a breakthrough for fluid-phase PK studies and confirmed the utility of ionspray MS for preclinical complex opioid profiling.

Dams et al. [9] developed and validated a direct-injection LC–APCI–MS/MS method for the simultaneous quantification of DIM, 6-MAM, morphine, NMOR and a wide panel of cocaine and methadone metabolites in human urine. The sample preparation was intentionally minimal: Urine (100 μL) was fortified with 10 μL internal standard solution, vortexed and centrifuged to remove debris. Subsequently, 10 μL supernatant was directly injected into the system—no SPE or derivatization required. Separation was achieved on a Synergi Polar RP column (150 × 2.0 mm, 4 μm, Phenomenex, Torrance, CA, USA) with a matching guard column (4 × 2 mm; Phenomenex, Torrance, CA, USA) using a 26-min gradient elution starting from 5% to 90% acetonitrile, with a mobile phase of 10 mM ammonium formate (pH 4.5) containing 0.001% formic acid. The system was interfaced with a Thermo Finnigan LCQ Deca XP ion-trap mass spectrometer with APCI in positive mode, and analytes were detected via selected reaction monitoring. Calibration was performed using deuterated internal standards and weighted linear regression (1/x), yielding excellent linearity (r^2^ > 0.99) up to 10,000 ng/mL. The method achieved LOQs of 10–100 ng/mL for all opioids, with an intra-/inter-day precision <16% and accuracy of 82.5–109.2%. Stability tests confirmed DIM susceptibility to hydrolysis at room temperature, necessitating cooled autosampler conditions (4 °C) to preserve analyte integrity. Although this method was not optimized for sub-nanogram sensitivity, it provided sufficient reliability and specificity for clinical drug monitoring applications, particularly for DIM and methadone compliance within the National Institute on Drug Abuse study framework.

LC-APCI-MS enables comprehensive DIM metabolite panels without derivatization or non-volatile salts, supporting rapid pharmacokinetic mapping in animal models and clinical matrices. Rodent studies show swift conversion of DIM to 6-MAM and morphine, followed by glucuronidation to M3G, with limited formation of M6G. Subsequent direct-injection LC-APCI-MS/MS assays in human urine minimized sample handling, expanded analyte coverage, and produced linear calibration (r^2^ > 0.99) with LOQs around 10–100 ng/mL. Although typically less sensitive than ESI-MS/MS, APCI-based workflows deliver stable performance with manageable matrix effects and are well suited to compliance testing and high-throughput forensic screening involving multiple drugs.

##### ESI–MS(/MS) with On-Line and Mixed-Mode SPE

Rentsch et al. [8] were among the first to apply C18 SPE coupled to LC–ESI–MS for human DIM pharmacokinetics. In parallel to the blood- and plasma-based investigations using a validated method described earlier by Bogusz et al. [47], Rentsch et al. [8] in 2001 developed and validated a versatile LC–MS protocol capable of quantifying morphine, 6-MAM, codeine and their glucuronides within autopsy blood, serum and other body fluids. SPE was performed using Bond Elut C18 cartridges (Varian, Harbor City, CA, USA), following conditioning and elution with methanol–acetic acid (9:1, *v*/*v*). The method yielded high recoveries (85–98%) across all analytes. Chromatographic separation was achieved on a Superspher RP-18 column (125 × 4 mm, 4 µm; Merck, Darmstadt, Germany), using an acetonitrile (J.T. Baker, Phillipsburg, NJ, USA)–ammonium formate buffer (pH 3.0, Merck, Darmstadt, Germany)). Time-scheduled SIM in positive ESI mode on a Hewlett-Packard LC–MS system (Waldbronn, Germany) tracked both native and deuterated species. Validation metrics in blood-based matrices yielded LOQs as low as 1 ng/mL for morphine and 2 ng/mL for 6-MAM, with r^2^ > 0.99 and CVs < 10%. Despite using the same protocol described by [47], Rentsch et al. [8] performed the analysis using a positive-ion ESI–multiple reaction monitoring mode (ESI-MRM) platform. LOQs reached ≤0.2 ng/mL for DIM and 6-MAM and ≤0.5 ng/mL for MOR, with recoveries above 85% and intra/inter-assay CVs < 10%. This workflow quantified rapid distribution (Tmax: 2–5 min) and clearance phases, correlating plasma profiles with the subjective drug “rush” and underpinning dose-adjustment guidelines in clinical DIM-assisted therapies.

Katagi et al. [18] developed a fully automated LC–ESI–MS method that incorporated on-line column-switching SPE Capcell Pak MF SCX 2 × 10 mm, 5 µm (Shiseido, Tokyo, Japan) for the rapid and sensitive identification of DIM, 6-MAM and morphine in human urine. Using a strong cation-exchange (SCX) cartridge to enrich cationic opiates directly from raw urine (50 µL), the system employed backflush elution onto a semi-micro SCX analytical column Capcell Pak MF SCX (2.0 × 10 mm, 5 µm; Shiseido, Japan) under isocratic conditions (10 mM ammonium acetate–acetonitrile, 30:70, *v*/*v*, pH 6) with ESI detection on a Micromass Platform quadrupole MS (Micromass Platform, Manchester, UK). Selected positive ion monitoring yielded limits of detection of 0.1 ng/mL (DIM), 0.5 ng/mL (6-MAM) and 3.0 ng/mL (MOR), with a linearity range of 1–100 ng/mL (DIM, 6-MAM) and 10–1000 ng/mL (morphine), recoveries > 94% and CVs < 6.7%. The method has been successfully applied in forensics cases to identify DIM and its markers alongside acetylcodeine and methamphetamine, confirming its utility for multiplexed drug exposure profiling in intoxication investigations.

Girardin et al. [40] advanced DIM pharmacokinetic profiling by implementing an online SPE–LC–ESI–MS workflow tailored for high-dose administration in patients with narcotics dependence, building directly on the validated protocols reported by Rentsch et al [8]. The assay utilized acidified plasma with optimized wash–elution conditions and a rapid 4-min gradient on a narrow-bore C18 column. This design achieved LOQs of 0.5 ng/mL for DIM and 0.2–0.5 ng/mL for its metabolites, with recoveries > 90% and intra/inter-assay precision (CVs) consistently <8%. Arterial plasma profiling following IV, intramuscular (IM) and oral dosing revealed linear kinetics up to 250 mg (IM) and 600 mg (oral), with oral administration yielding negligible systemic DIM and 6-MAM concentrations but unexpectedly high morphine bioavailability (67% ± 19%) and glucuronide output (M3G: 205% ± 52%; M6G: 180% ± 61%). Importantly, case sample data exposed modality-specific metabolic shifts: IM dosing prolonged DIM exposure and delayed 6-MAM peaks, while oral administration elicited rapid morphine absorption and elevated M6G/MOR ratios. Therefore, these metrics can be leveraged to stratify overdose risk and guide safe dose adjustment in clinical DIM-assisted therapies.

Rook et al. [20,32] established a benchmark LC–ESI–MS/MS platform for DIM pharmacokinetics by integrating mixed-mode SPE (Oasis MCX cartridges, Waters Corp., Milford, MA, USA) with rapid, high-sensitivity chromatographic separation. In their 2005 study [32], they developed a 5-min assay capable of quantifying DIM, 6-MAM, morphine and its glucuronides (M3G, M6G) (Cerilliant, Austin, TX, USA) in plasma using only a 0.25-mL sample. Plasma was acidified with 0.15 N HCl (VWR International, Leuven, Belgium) and applied to Oasis MCX SPE cartridges. After loading, cartridges were washed with 0.1% acetic acid and water, then eluted with 0.5% ammonium acetate in methanol (*v*/*v* 1:20) into 50 mM ammonium acetate buffer (pH 3.0) to stabilize DIM. The eluate was evaporated under nitrogen, reconstituted and injected into a 50 × 2.1 mm C18 column; Agilent Technologies, Palo Alto, CA, USA). A rapid gradient from 5 mM ammonium acetate (pH 4.0, Fluka, Buchs, Switzerland) to acetonitrile at 0.5 mL/min (Merck, Darmstadt, Germany) achieved baseline separation of M3G and M6G despite shared MRM transitions, and the total run time was 5 min. Using deuterated internal standards and positive-ion MRM on a triple quadrupole, the method achieved a linearity range of 5–500 ng/mL for all analytes (r^2^ > 0.998). Recoveries ranged 79.8–86.6% for DIM and 76–101% for metabolites. Intra-assay precision and accuracy at the LLOQ (5 ng/mL) achieved ≤20% CV and ≤15% CV at higher levels for DIM and its metabolites, respectively. No significant ion suppression or endogenous interferences were observed. This high-throughput, high-sensitivity platform enables full-panel quantification of DIM and its metabolites in a single, rapid assay ideal for clinical toxicology, PK studies and therapeutic monitoring.

Rook et al. [20] further extended their rapid LC–ESI–MS/MS platform to characterize DIM pharmacokinetics under dose-escalation and steady-state conditions in patients with opioid dependence. Plasma (0.25 mL) was processed via MCX SPE and analyzed on a 50 × 2.1 mm C18 column with a 7-min gradient. LOQs reached 0.05 ng/mL for DIM, 6-MAM, morphine, M3G and M6G. Recoveries for all analytes exceeded 85%, and both intra- and inter-assay precision remained below 10% CV. By sampling across 67–150% of each subject’s maintenance DIM dose over consecutive administrations, the authors demonstrated linear dose–AUC relationships for DIM, 6-MAM and morphine but observed non-linear accumulation of M3G and M6G during steady-state dosing. The disproportionate increase in glucuronide exposure indicated metabolic induction with repeated DIM administration, underscoring the need to monitor inactive metabolite buildup in long-term DIM maintenance. Altogether, these findings established the MCX SPE–LC–MS/MS assay as a high-throughput, high-sensitivity tool for both acute and chronic DIM PK studies.

Klous et al. [19] developed and applied a validated LC–MS/MS method to compare the pharmacokinetics of smoked DIM using two different inhalation techniques: chasing the dragon versus a temperature-controlled heating device. Plasma samples were obtained from five male patients with DIM dependence after inhalation of 50 mg DIM/caffeine tablets by alternating between the two methods over 5 consecutive days. Sample preparation involved SPE using Oasis MCX cartridges (Oasis MCX cartridges, Waters Corp., Milford, MA, USA) following acidification and centrifugation. Subsequently, quantification was performed using an HPLC system coupled to a triple-quadrupole MS/MS (Waters Quattro Premier XE) equipped with an XTerra MS C18 column (150 × 2.1 mm, 3.5 µm, Waters Corporation, Milford, MA, USA). The mobile phase consisted of ammonium formate buffer and acetonitrile, applied in a gradient previously validated for urine analysis. LLOQs for all analytes (DIM, 6-MAM, morphine, M3G and M6G) were 5 ng/mL, with inter-assay accuracy < 5% and precision < 11%. Findings revealed significantly lower systemic exposure (AUC) and peak plasma concentrations (Cmax) for DIM and 6-MAM at 80% and 73%, respectively, when using the heating device than when chasing the dragon. Morphine and its glucuronides also showed reduced exposure (~38–42%). Despite equivalent elimination t½, these results confirm that chasing the dragon is a more efficient method for DIM vaporization and systemic delivery than standardized heating, likely due to greater user control over liquid movement, inhalation timing and chasing the dragon efficiency.

Karinen et al. [43] developed a highly sensitive and selective LC–ESI–MS/MS method for the simultaneous quantification of DIM, 6-MAM, morphine and their glucuronide conjugates (M3G and M6G) in small-volume whole blood samples (100 µL) and mouse brain tissue. To stabilize the labile acetylated analytes, sodium fluoride and acidic ammonium formate buffer (pH 3.1) were added immediately post-collection, followed by protein precipitation using ice-cold acetonitrile–methanol (85:15). After centrifugation and evaporation to dryness, samples were reconstituted with cold mobile phase and injected into a Waters Quattro Premier XE LC–MS/MS system (Waters Micromass, Manchester, UK). Chromatographic separation was performed on an XTerra MS C18 column (150 × 2.1 mm, 3.5 µm, Waters Corporation, Milford, MA, USA) with a 16-min gradient (3–60% acetonitrile). MRM detection achieved LOQs of 0.33–2.5 ng/mL in blood and 2.0–7.7 ng/g in brain tissue, with recoveries of 80–111% and CVs of 2.1–14.5%. The method exhibited excellent linearity (r^2^ > 0.995) and negligible matrix effects across biological matrices. Using samples from DIM-treated mice, the assay confirmed greater analyte stability in brain tissue than in blood and enabled the simultaneous monitoring of parent drug and metabolites with minimal sample volume and cleanup, marking a major advancement in rodent-based PK and neurotoxicity research.

Al-Asmari et al. [1] investigated pediatric DIM analgesia by developing an LC–ESI–MS/MS assay (Thermo Finnigan LCQ DECA XP Plus ion-trap MS (San Jose, CA, USA) with Surveyor LC system interface, ESI positive-ion mode) using 0.25-mL plasma samples collected from children receiving IV DIM or IN DIM. Following addition of deuterated DIM, 6-MAM, morphine, M3G, M6G and NMOR internal standards, plasma was loaded onto Bond Elut C18 SPE cartridges (Varian, Palo Alto, CA, USA), washed with 5 mM ammonium carbonate (pH 9.3)–water and eluted with methanol. Separation was performed on a Synergy Polar RP column (150 × 2.0 mm, 4-μm particle size, Phenomenex, Torrance, CA, USA) using a 36-min gradient of 10 mM ammonium formate (pH 3)–acetonitrile at 0.3 mL/minute prior to ESI–SRM quantification. Calibration was linear with a range of 0.1–50 ng/mL (r^2^ > 0.999); LOQs ranged 0.28–1.22 ng/mL; LODs were 0.08–0.37 ng/mL; recoveries were 81–109%; and intra-/inter-day precision ranged 2.5–13.4% and 1.8–15%. Case findings revealed that IV DIM more rapidly produced and higher plasma concentrations of DIM and 6-MAM, peaking within 5 min, while IN DIM resulted in delayed and lower peak levels, with 6-MAM and MOR reaching therapeutic thresholds by 10–15 min. Despite reduced bioavailability, IN DIM still achieved effective analgesic metabolite exposure, demonstrating its value as a non-invasive alternative for pediatric pain management.

Moreno-Vicente et al. [45] developed and validated a high-throughput LC–MS/MS method for simultaneously quantifying DIM, its metabolites (6-MAM, morphine, M3G, M6G), naloxone (NAL) and qualitatively detecting naltrexone (NALT) in human plasma. Using minimal sample volume (100 µL), the method incorporated protein precipitation with cold methanol (Merck, Darmstadt, Germany), a dual-step reconstitution strategy, and separation on a phenyl column (XBridge Phenyl column (150 × 4.6 mm, 5 µm; Waters Corporation, Milford, MA, USA) with 10 mM ammonium formate–formic acid (pH 3.0; Sigma-Aldrich, St. Louis, MO, USA) and acetonitrile (Merck, Darmstadt, Germany) gradient. The analytical platform demonstrated excellent sensitivity (LOQs: 10–2000 ng/mL), near-complete recoveries (~100%), linearity (r^2^ > 0.995) and precision (<11.5% CV). Applied in a clinical trial involving oral administration of DAM/NAL tablets (50/2 mg), the assay completely identified first-pass metabolism of DIM and 6-MAM, with morphine and its glucuronides serving as key circulating markers. M3G showed the highest plasma exposure Area under the curve (AUC)_inf: 5564.5 ng·h/mL), confirming its role as the predominant inactive metabolite, while NALT was sporadically detected. This validated, Good Laboratory Practice-compliant method offers a robust solution for pharmacokinetic and bioequivalence studies involving DIM-opioid antagonist combinations.

Gottas et al. [7] extended LC–ESI–MS/MS DIM analysis to postmortem vitreous humor and paired blood, preserving labile acetyl esters and glucuronides with a streamlined protein-precipitation/liquid–liquid extraction. For vitreous humor, 100-μL aliquots were spiked with 50 μL deuterated internal standard solution (0.5 μM in water, Lipomed GmbH, Arlesheim, Switzerland), then extracted with 500 μL acetonitrile–methanol (85:15, *v*/*v*). Blood samples (200 μL) were spiked the same 50 μL internal standard before protein precipitation and underwent an identical acetonitrile–methanol extraction process. After centrifugation, supernatants were evaporated, reconstituted in the mobile phase and injected into a Waters Quattro Premier XE triple-quadrupole mass spectrometer coupled to an HPLC system (XE MS/MS) equipped with an XTerra MS C_18_ column (150 × 2.1 mm, 3.5 μm, Waters Corporation, Milford, MA, USA). Chromatographic conditions mirrored their validated urine method, an aqueous ammonium formate–acetonitrile gradient at 0.3 mL/min, yielding sharp peak shapes for DIM, 6-MAM, morphine and M3G and M6G. Multiple-reaction-monitoring transitions produced LOQs of 0.2–1 ng/mL, recoveries of 80–95% and intra-/inter-assay precision < 12% CV. This approach reliably captured the fleeting 6-MAM marker in vitreous humor up to 12 h post-mortem and provided concordant metabolite profiles in blood, reinforcing its utility for forensic confirmation of recent DIM use.

LC–ESI–MS(/MS) has become the leading approach for DIM bioanalysis because it ionizes polar metabolites, including glucuronides, and supports broad multi-analyte panels across blood, urine, saliva, and tissue. With isotope-dilution, mixed-mode SPE, and MRM acquisition, validated methods routinely reach sub-ng/mL limits with high linearity (r^2^ > 0.995) and robust precision. These capabilities enable simultaneous quantification of DIM, 6-MAM, morphine, M3G, and M6G for pharmacokinetic and pharmacodynamic studies, therapeutic monitoring, and forensic applications. Although matrix effects and ion suppression require careful control, LC–ESI–MS(/MS) remains a reference platform for opioid profiling, combining sensitivity, coverage, and practicality.

### 3.2. Sample Matrices and Sampling Considerations

The unstable nature of DIM, which is characterized by rapid enzymatic and chemical deacetylation, requires specimen choice and handling that is as critical as analytical platform selection [9,38]. Across forensic, clinical and preclinical contexts, each matrix demands trade-offs between detection window, metabolite stability and collection or storage logistics [8,18].

Hair remains the definitive matrix for long-term DIM exposure, sequestering DIM and its metabolites within keratin over weeks to months [15,44]. Typical workflows collect 50–100 mg of proximal strands, apply rigorous decontamination (detergent followed by methanol or organic-solvent washes), then disrupt the matrix by pulverization or sonication. Extraction is performed using basified methanol or SPE, and analytes are derivatized (MBTFA or TMCS) before GC–MS. These methods achieve low LOQs (as low as 0.02 ng/mg hair), with recovery rates of 55–99% and precision < 9% CV. Despite its unrivaled retrospective window, hair analysis is hindered by external contamination (sweat, smoke residues), in-process hydrolysis and diminished sensitivity for detecting very recent or low-dose use [48].

Sweat patches capture cumulative excretion over 24–72 h with low local esterase activity, enabling DIM capture. However, although sweat patches are excellent for compliance monitoring, extraction is affected by variable perspiration rates, weak dose–concentration relationships, adhesive interferences and in-patch conversion of DIM to 6-MAM [16,17].

Blood and plasma are the definitive matrices for acute use, enabling DIM and 6-MAM detection within minutes, which is ideal for correlating pharmacokinetics with clinical effects. Analysis involves cold-chain collection, acidification, rapid centrifugation and sometimes derivatization to prevent ex vivo hydrolysis [8,19,32].

Urine, enriched in morphine glucuronides (M3G, M6G), offers an 8- to 24-h detection window with minimal cleanup (dilution or SPE). However, the indirect detection of the parent drug limits forensic specificity [9,18]. Oral fluid (saliva) can accurately indicate free plasma levels soon after dosing but demands rigorous mouth-rinse protocols to avoid contamination [43].

Postmortem fluids, such as vitreous humor and cerebrospinal fluid, have low esterase activity that enables prolonged postmortem detectability of fleeting markers like 6-MAM. Furthermore, these fluids help distinguish recent drug use from postmortem redistribution in central blood. In contrast, solid tissues (e.g., brain, liver) may retain unconjugated DIM for longer intervals but require intensive homogenization and cleanup to mitigate matrix interference [7,47].

Practical sample logistics vary across analytical methods. Plasma LC–MS(/MS) assays typically consume <0.5 mL, whereas hair analysis may need ≥50 mg. Fluids must be aliquoted, acidified (approximately pH 4) and frozen (≤−20 °C) to inhibit hydrolysis, while derivatized GC–MS extracts require storage at 4 °C in amber vials under inert gas. Notably, dried blood spots offer intrinsic long-term stability [1,8,46].

### 3.3. Stability of DIM

Despite decades of research, the stability of DIM remains inherently questionable, with few studies offering robust experimental frameworks for assessing its degradation in human matrices. Numerous investigations have highlighted significant hydrolysis during extraction and storage, particularly in aqueous or methanolic environments. For instance, Wijesekera et al. [49] demonstrated the rapid degradation of DIM in methanolic solution, incurring a 90.8% loss rate at ambient temperature and 70% under refrigeration. Complete conversion to 6-MAM occurred after 8 weeks at room temperature. In contrast, DIM in solid powder form exhibited greater resilience, with degradation limited to <21% at 26 °C and <17% at 6 –8 °C, underscoring the matrix-dependent nature of DIM’s chemical integrity.

To circumvent these limitations, alternative matrices, such as hair and sweat, have been explored, given their presumed stability and minimal post-collection handling requirements. Hair is a promising matrix because its analysis eliminates cold-chain logistics and rapid-freezing protocols. Goldberger et al. [15] found that the DIM content of human hair remains chemically stable during extraction, with <10% conversion to 6-MAM. Further optimization using acetonitrile for both standard preparation and SPE wash steps reduced degradation to <5%, while LC–MS analysis of reconstituted extracts confirmed analyte stability in the autosampler, preserving DIM, 6-MAM and MOR levels.

Musshoff et al. [27] conducted a targeted evaluation of DIM stability in hair extracts under controlled conditions. In this study, hair specimens spiked with DIM, 6-MAM and morphine were extracted and analyzed at multiple time points (0, 1, 3, 12, 18 and 24 h) while stored at room temperature in an autosampler. They found that DIM remained chemically stable throughout the entire 24-h period, with no measurable degradation to 6-MAM or morphine. This finding reinforces the suitability of hair as a robust matrix for DIM detection, particularly in forensic workflows requiring delayed analysis or ambient storage. The study also supports the reliability of segmental hair testing for retrospective drug exposure, providing close adherence to validated extraction protocols is observed.

Kintz et al. [17] conducted one of the earliest controlled studies on DIM stability in sweat, using sweat patches to collect samples from 14 subjects undergoing IV DIM maintenance therapy. The patches were worn for approximately 24 h and stored at −20 °C prior to analysis. In this study, DIM was the predominant analyte in sweat, with minimal conversion to 6-MAM and morphine, indicating that the parent compound was chemically stable under the analytical conditions. Brunet et al. [42] expanded on this by validating an SPE–GC–MS method for the simultaneous quantification of DIM and its metabolites in sweat. Their protocol, which included sodium acetate buffer extraction and derivatization, demonstrated that DIM hydrolysis during specimen processing was consistently <11%, confirming that the DIM content of sweat—when properly collected and stored—can be preserved with acceptable analytical integrity. Together, these studies demonstrate sweat as a viable matrix for DIM monitoring, provided that standardized collection and extraction protocols are followed.

In biofluid applications, methodological refinements have focused on minimizing hydrolysis through chemical stabilization and internal standardization. Wang et al. [36] introduced sodium fluoride as a preservative during sample processing, achieving <10% degradation. Guillot et al. [37] further demonstrated that DIM, 6-MAM and MOR remained chemically intact for several weeks when reconstituted in chloroform containing 0.1% pyridine and stored at ambient temperature, reinforcing the importance of solvent composition in post-extraction stability.

Zuccaro et al. [38] quantified DIM and its metabolites in serum following subcutaneous administration. Despite a reported LOQ for DIM of as low as 0.5 ng/mL, the parent compound was entirely undetectable in all biological samples, including those collected immediately post-injection. The authors did not experimentally investigate DIM stability or implement stabilization protocols, such as cold-chain handling, buffering or enzymatic inhibition. Given the absence of DIM and the presence of its downstream metabolites, particularly 6-MAM and morphine, the findings strongly indicate complete in vivo or procedural hydrolysis of DIM prior to or during sample processing. The lack of controlled extraction timing, cold-storage protocols or sample-specific hydrolysis mitigation strategies likely contributed to the failure in detecting intact DIM. This contrasts with the results of Bourquin et al. [31], who successfully preserved and detected DIM in human samples using LC–ESI–MS and optimized C8 SPE cartridges, underscoring the importance of validated stabilization procedures when targeting DIM in biological fluids.

Katagi et al. [18] investigated DIM detection in urine samples collected immediately after injection from active users and successfully detected the parent compound. However, the study did not perform any controlled assessments of DIM stability during extraction or post-collection handling, leaving open the possibility of unquantified degradation. In contrast, Dams et al. [9] conducted targeted stability evaluation of DIM in urine and demonstrated that the compound was chemically unstable and underwent significant degradation when stored at room temperature for 24 h. Notably, DIM remained intact when samples were refrigerated at 4 °C for 24 h, frozen at −20 °C for 5 days, or subjected to five freeze–thaw cycles. The authors attributed the reduced hydrolysis rate to the naturally acidic pH of urine, which may offer partial protection against enzymatic or spontaneous breakdown. Altogether, these studies underscore the importance of temperature control and matrix-specific conditions when preserving DIM in biological fluids.

Capella-Peiró et al. [41] evaluated DIM and metabolite detection in serum using MLC with direct injection but also did not incorporate any formal DIM stability assessment. In this study, DIM was not detected in any of the real-case serum samples, although 6-MAM and morphine were consistently found. The absence of DIM was attributed to uncontrolled sample collection and handling, which likely allowed for complete hydrolysis prior to analysis. No stabilizing agents, temperature control or timed extraction protocols were applied, and the authors did not investigate degradation kinetics or matrix-specific preservation strategies. Consequently, the study underscores the critical importance of validated stabilization procedures, such as cold-chain storage, pH buffering and enzymatic inhibition, when attempting to quantify DIM in serum, particularly in forensic or clinical settings where sample integrity is time-sensitive.

Rook et al. [32] conducted a comprehensive stability evaluation of DIM and its metabolites in plasma and stock solutions. They found that DIM, 6-MAM, morphine and their glucuronide conjugates (M3G, M6G) remained chemically stable under multiple conditions. Short term integrity was confirmed via a 60-min ice/water bath, while refrigerated storage at 4 °C preserved analyte concentrations for up to 17 days. Long-term stability was maintained in frozen stock solutions at −20 °C for 25 months and in patient plasma samples stored at −20 °C for 14 months. Furthermore, no degradation was observed following three freeze–thaw cycles. The authors emphasized that degradation was effectively mitigated by the use of acidified plasma post-thaw and a mobile phase buffered at pH 4 during chromatographic separation, demonstrating the importance of pH control and temperature management in preserving DIM and its metabolites for delayed analysis.

Al-Asmari et al. [1] conducted a targeted stability assessment of DIM and its metabolites in pediatric plasma samples using a validated LC–MS/MS method optimized for low-volume specimens. Recognizing the vulnerability of DIM to hydrolysis during sample preparation, the study incorporated Bond Elut C18 SPE, deuterated internal standards and ESI ionization to minimize degradation. Stability was rigorously evaluated across multiple conditions: DIM remained chemically intact for 24 h in the autosampler, underwent <5% degradation during refrigeration for 72 h, and retained >98% recovery after 30 days of storage at −20 °C. Furthermore, freeze–thaw resilience was confirmed over five cycles, with a cumulative loss rate < 5%. These findings affirm that, with appropriate stabilization protocols, DIM and its metabolites can be reliably quantified in pediatric plasma, even at low therapeutic concentrations and limited sample volumes.

The stability of DIM clearly hinges on handling conditions rather than any inherent chemical robustness. In calibrated stock solutions and aqueous extracts, DIM can hydrolyze almost completely within weeks unless stringent measures for cold-chain handling, pH control or stabilization with agents, such as sodium fluoride, are employed. Conversely, solid or lyophilized formulations, as well as matrices such as hair and sweat, require far less aggressive preservation strategies, maintaining DIM integrity with minimal loss. Biofluids, however, demand a tailored combination of refrigerated or frozen storage, the use of acidified buffers or optimized SPE solvents (acetonitrile or chloroform–pyridine) and isotopically labeled internal standards to suppress artifactual breakdown. When these measures are implemented alongside validated autosampler protocols and low-pH chromatographic conditions, DIM, 6-MAM and morphine remain unaffected by storage or repeated freeze–thaw cycles.

Reliable quantification of DIM depends on stabilization strategies that complement the chosen matrix. For retrospective screening, hair and sweat provide operational simplicity without elaborate cold-chain logistics. For plasma, urine or serum, strict temperature management, pH buffering, enzymatic inhibitors and deuterated standards are essential. By harmonizing these matrix-specific protocols, validating them under routine laboratory conditions, and supplementing them with inter-laboratory proficiency testing, clinical and forensic toxicology teams can achieve consistent, reproducible measurements of DIM and its metabolites.

## 4. Discussion

Over the past three decades, the analytical toxicology of DIM and its early metabolites have matured into a diversified toolkit that spans more than five orders of magnitude in sensitivity and accommodates a spectrum of forensic, clinical and research needs. On one extreme, derivatized GC–MS assays can achieve sub-pg/mg limits in hair [48] and low-pg/mL detection in sweat [16,42], but impose extended sample preparation workflows (SPE/LLE, TMS/BSTFA or MBTFA derivatization). On the other end, HPLC–UV/FL and micellar LC–UV methods deliver rapid 2- to 10-min run times with minimal sample preparation suitable for routine monitoring of morphine and its glucuronides at tens of ng/mL sensitivity [30,41]. Nestled between these extremes, modern LC–MS(/MS) platforms strike the optimal balance: sub-nanogram sensitivity (as low as 0.05 ng/mL; [20]), 3–10 min runs, moderate SPE or protein-precipitation preparation and multiplexed quantification of DIM, 6-MAM, morphine, M3G, and M6G [19,45]. Although GC–MS and HPLC–UV/FL methods were historically central to DIM analysis, their limitations have become increasingly apparent. GC–MS often required labor-intensive derivatization and offered limited sensitivity for polar metabolites, while HPLC–UV/FL lacked the selectivity to resolve structurally similar compounds at low concentrations. These constraints, combined with the growing demand for multi-analyte panels and sub-ng/mL detection, have led to a gradual shift toward LC–MS(/MS) platforms. As summarized in Table 1, LC–MS(/MS) provides superior sensitivity, specificity, and throughput, making earlier methods less suitable for current forensic and clinical applications.

### 4.1. Sensitivity vs. Throughput

Derivatized GC–MS remains unrivalled in hair and sweat analysis when ultra-trace detection is non-negotiable; however, throughput suffers from lengthy extraction and derivatization steps [2,15]. Conversely, UV-based HPLC/MLC assays excel in high-throughput settings involving the processing of hundreds of urine or plasma samples per day. However, this approach has limited ability to detect short-lived or low-dose markers [1,32]. LC–MS(/MS) methods effectively bridge this gap, delivering sub-ng/mL limits with throughputs comparable to HPLC–UV, thanks to on-line SPE automation and fast gradients [18,30,31].

### 4.2. Hydrolysis Control

DIM and 6-MAM are notoriously labile, with ex vivo deacetylation artifactually inflating downstream metabolite levels. Robust LC–MS(/MS) workflows now incorporate cold-chain sampling, rapid protein precipitation, acetonitrile SPE washes and even room temperature derivatization to preserve parent analytes [20,32]. In contrast, high-temperature GC–MS derivatization or direct-injection APCI approaches that omit ester protection risk underestimating DIM and overestimating morphine [1,38].

### 4.3. Matrix Compatibility

The choice of biological matrix strongly influences both the analytical platform and the achievable sensitivity. Keratinized matrices such as hair require GC–MS with rigorous decontamination and cleanup to overcome external contamination and achieve ultra-trace detection limits [16,42,50]. Sweat, typically collected via patches, also favors GC–MS due to the very low analyte abundance and the need for robust cleanup to avoid environmental interferences [25,30].

In contrast, blood, plasma, and urine are more versatile: abundant glucuronide conjugates can be rapidly quantified by HPLC–UV/FL, whereas LC–MS(/MS) with mixed-mode or C18 SPE is preferred for comprehensive multi-analyte panels and sub-ng/mL detection, tissue and vitreous humor samples, often encountered in postmortem casework, are best analyzed by LC–MS(/MS). These matrices demand careful interpretation due to postmortem redistribution and heterogeneous composition [5,18]. Oral fluid (saliva) presents unique challenges, as pre-collection rinsing is essential to minimize oral cavity residues, and ensure analytical specificity [43]. A comparative overview of matrix–platform compatibility, challenges, and typical sensitivity ranges is provided in Table 2.

### 4.4. Metabolite Coverage

Early GC–MS hair assays targeted only DIM, 6-MAM and morphine [13,30]. Current LC–MS(/MS) methods routinely quantify up to seven analytes, including M3G, M6G and opioid antagonists, in one injection [1,18,43]. Such comprehensive panels improve forensic interpretation by enabling survival time estimation via parent-to-metabolite ratios and monitoring glucuronide accumulation during maintenance therapy.

### 4.5. Automation and Emerging Technologies

On-line SPE column-switching [18], ultra-fast UPLC gradients and high-resolution accurate-mass systems (Orbitrap, time of flight) are rapidly gaining traction [21,22,23]. These innovations promise sub-2-min analyses, enhanced isobaric discrimination (e.g., codeine vs. morphine) and non-targeted screening of novel synthetic opioids and adulterants.

### 4.6. Limitations and Future Work

This review is narrative in design and therefore subject to inherent limitations, including potential selection bias and lack of standardized synthesis. The included studies exhibit considerable heterogeneity in terms of analytical platforms, sample preparation protocols, and validation parameters, which complicates direct comparison. Furthermore, no quantitative meta-analysis was performed due to variability in reporting formats and insufficient statistical harmonization across studies. Notably, several papers reporting important findings lack basic validation parameters, such as recovery, precision, or matrix effects, which limits their forensic applicability. This may reflect the fact that some of these studies were primarily designed for clinical investigation, where method validation guidelines are not uniformly practiced or mandated for publication. These limitations should be considered when interpreting the comparative findings and recommendations presented.

Analytical methodologies for DIM detection have evolved from basic single-analyte, derivatized assays into sophisticated, high-throughput platforms capable of tracking the entire metabolic cascade with sub-nanogram precision. Innovations in sample stabilization, hydrolysis control, mixed-mode extraction and automated on-line cleanup have significantly preserved the stability of labile acetyl esters, enabling more accurate quantitation across blood, urine, hair, sweat and postmortem matrices. Cutting-edge technologies, such as high-resolution accurate-mass spectrometry, ultrafast LC and ion-mobility separation, are set to enhance specificity and identify novel biomarkers. The integration of machine learning for spectral deconvolution and retention time prediction promises to streamline method development and elevate interpretative depth. These advancements will empower toxicologists to differentiate recent use from historical exposure, monitor compliance, detect emerging adulterants and clarify complex postmortem presentations. Continued progress in instrumentation and sample processing workflows remains critical to strengthening forensic, clinical and public health responses in the evolving landscape of opioid use.

## 5. Conclusions

This review traced the analytical evolution of DIM analysis, from early GC–MS assays to modern high-throughput LC–MS/MS platforms, highlighting the critical role of these methodologies in advancing forensic toxicology, clinical monitoring, addiction medicine and emergency care. The most recent validated method, reported in 2021, reflects the remarkable technological progress that now enables sub-nanogram sensitivity, full metabolic profiling and rapid turnaround times. Nevertheless, this journey is far from complete. Future studies should expand beyond conventional matrices and explore underutilized specimens, such as hair, sweat, saliva, gastric contents and stomach tissue, especially in postmortem contexts where DIM biomarkers may persist and offer enhanced forensic interpretability.

## Figures and Tables

**Figure 1 toxics-13-00867-f001:**
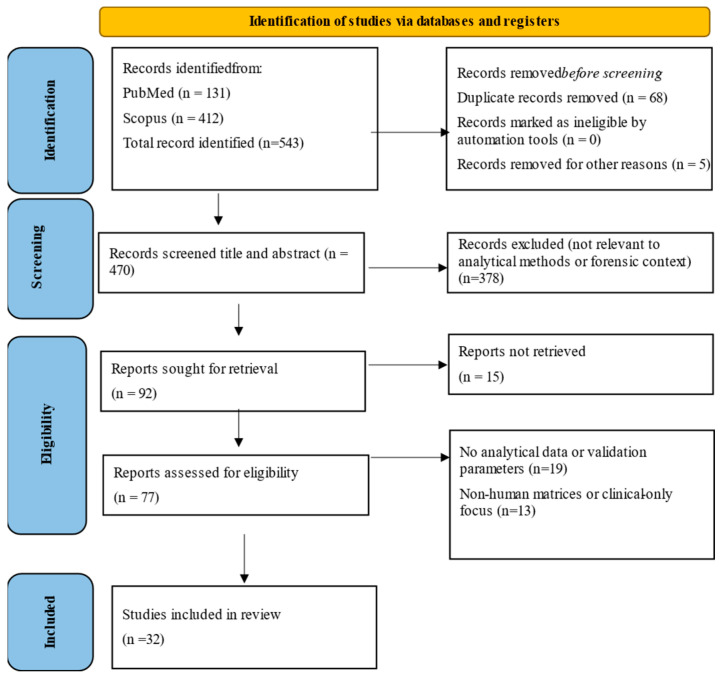
PRISMA 2020 flow diagram of the study selection process for diamorphine (DIM) quantification methods (initial pool = 543).

**Table 2 toxics-13-00867-t002:** Matrix Compatibility of Analytical Platforms for DIM and Metabolites.

Matrix	Key Considerations	Typical Sensitivity (LOD/LOQ)
Hair	Requires extensive washing and decontamination; keratin binding complicates extraction	Ultra-trace (pg–ng/mg)
Sweat	Patch collection; low analyte abundance; risk of environmental contamination	ng/patch
Blood/Plasma/Urine	Matrix effects from proteins and salts; need for SPE cleanup	Sub-ng/mL achievable
Oral Fluid (Saliva)	Pre-collection rinsing needed; risk of oral cavity contamination	Low ng/mL
Tissues/Vitreous Humor	Postmortem redistribution; heterogeneous composition	ng/g or ng/mL

## Data Availability

Whole data is included in the manuscript and will be available online.

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
