# Peer review of "Analytical Methods for the Determination of Diamorphine (Heroin) in Biological Matrices: A Review"

_toxics, 2025, doi:10.3390/toxics13100867_

Round 1

Reviewer 1 Report

Comments and Suggestions for Authors

The manuscript provides a narrative review of analytical methods used to detect diamorphine and its metabolites in biological matrices, covering studies published between 1980 and 2025. It offers a broad overview of the application of HPLC with optical detectors, GC-MS, and LC-MS(/MS) for measuring diamorphine, 6-MAM, morphine, and their glucuronides in various samples such as blood, urine, saliva, hair, sweat, and vitreous humor. The strength of the paper lies in the attempt to categorize analytical platforms according to context, which helps readers understand how different techniques have historically been employed in both clinical and forensic settings.

However, the methodological approach must be more transparent. The authors describe performing searches in PubMed and Scopus, yet they do not provide the exact search terms, the number of records initially retrieved, or how duplicates and exclusions were handled. The inclusion and exclusion criteria are only briefly mentioned and not systematically applied or documented. A PRISMA flow diagram, complete with numbers at each stage, would resolve these ambiguities and make the methodology more rigorous. This should be figure 1. 

Another area of weakness is in the way comparisons between methods are handled. The review discusses the performance of HPLC, GC-MS, and LC-MS(/MS), but the comparisons remain largely qualitative. LOD and LOQ values are reported inconsistently, often in different units, making it impossible to fairly compare performance across techniques. In addition, results from heterogeneous biological matrices are pooled together, although each matrix poses unique challenges and requires distinct analytical approaches since they have different detection windows. Separating the discussion and tables by matrix would significantly improve clarity and allow for a more meaningful evaluation of which technique is best suited to each type of sample. 

Moreover, while the authors do discuss sample preparation in general terms (for example, LLE, SPE, and in some cases derivatization), they do not go into detail about additional extraction or hydrolysis steps that are often crucial for certain metabolites. In particular, the use of a water bath hydrolysis step for urine samples to liberate glucuronides (e.g., morphine-3-glucuronide and morphine-6-glucuronide) is a well-established procedure in forensic toxicology, but the review does not highlight or critically evaluate this.

Hydrolysis and stability issues are acknowledged but not explored in sufficient depth. Since the breakdown of diamorphine into 6-MAM and morphine is a central problem in forensic toxicology, the review should dedicate more attention to how different methods address this challenge, for example through the use of additives, temperature control, or rapid extraction protocols. 

Overall, the review covers an important and interesting topic and brings together decades of research on diamorphine analysis. To increase its impact, however, the author should strengthen the methodological transparency of their review process, standardize and stratify comparisons, emphasize stability issues, and improve the interpretive value of their presentation.

As a final recommendation, the addition of a graphical abstract would be highly beneficial. A visual summary of the different analytical techniques, their relative strengths and weaknesses, and their suitability across biological matrices would enhance accessibility and make the paper more engaging for both specialized and non-specialized readers.

Comments on the Quality of English Language

The manuscript is generally understandable, but the English expression requires improvement to meet publication standards. At times, the writing is repetitive and overly descriptive, with long sentences that obscure the main point. There are also occasional inconsistencies in terminology (e.g., switching between different units or abbreviations without explanation). 

Author Response

We sincerely thank Reviewer 1 for their thoughtful and constructive feedback. We have carefully considered each comment and revised the manuscript accordingly to enhance its clarity, rigor, and scientific value. Please check our detailed responses in the attached 

Reviewer 2 Report

Comments and Suggestions for Authors

Please see the attachment file

Author Response

We thank the reviewers for their thoughtful and constructive feedback. We have carefully addressed each point and revised the manuscript accordingly. Please note that several line and page references cited in the review do not align with the final layout of our manuscript due to formatting changes during revision. Nevertheless, we have interpreted the intent of each comment and ensured that all suggested improvements have been implemented in the appropriate sections. Please check our detailed responses in the attached 

Round 2

Reviewer 1 Report

Comments and Suggestions for Authors

The author have addressed all of my previous concerns. I have only one minor comment: Table S1 contains important information that strengthens the study. Could the authors clarify why it has been placed in the supplementary information rather than the main manuscript? I believe it would add more value if included in the main text. In addition, the references listed in Table S1 should also be incorporated into the main manuscript. In my opinion, supplementary material is not necessary for this paper.

Reviewer 2 Report

Comments and Suggestions for Authors

Thank you for your thorough revisions. The manuscript is now clear, well-structured, and methodologically robust. The improvements to the abstract, methods, discussion, and the addition of a limitations section and comparative figure have significantly enhanced its value. I am satisfied with the current version and have no further suggestions.
